# Molecular and immunological heterogeneity of eosinophilic esophagitis: Insights and subtyping

Eric Twum[1], Ancha Baranova[1,2], Aman Ullah[1]*

**1** School of Systems Biology, George Mason University, Fairfax, Virginia, United States of America,
**2** Research Centre for Medical Genetics, Moscow, Russia

* aullah3@gmu.edu

## Abstract

### Background

Eosinophilic esophagitis (EoE) and its related subtypes—such as EoE-like esophagitis, lymphocytic esophagitis, and nonspecific esophagitis—pose significant diagnostic challenges due to overlapping clinical, histological, and endoscopic features. Although conventional EoE is well-characterized as a Th2-mediated disorder, the molecular and immunological drivers for its subtypes are poorly understood. We aimed to elucidate the unique molecular signatures underlying these esophageal inflammatory conditions, with the goal of refining disease classification and paving way to targeted therapeutic approaches.

### Methods

We performed an integrative multi-omics analysis incorporating differential gene expression profiling, weighted gene co-expression network analysis (WGCNA), functional enrichment studies, and machine-learning algorithms to identify molecular hallmarks that differentiate EoE from its subtypes. After examining subtype specific alteration in immune and metabolic pathways, novel biomarkers and regulatory mechanisms were uncovered.

### Results

Conventional EoE exhibited a distinct upregulation of periostin (*POSTN*), reinforcing extracellular matrix remodeling as its primary pathogenic mechanism. We identified *DNAH11* as a key player in epithelial turnover and esophageal dysmotility, revealing its previously unrecognized role in EoE pathogenesis. Suppressed zinc-related pathways (*MT1X, MT1F, MT2A*) suggest epithelial barrier dysfunction, with differential zinc transporter expression (*SLC39A1, SLC39A2*) indicating disruptions in zinc homeostasis, which may have therapeutic implications. Additionally, aberrant *CDX2* expression linked to methyl-CpG binding proteins suggests an epigenetic contribution

**Data availability statement:** All relevant data are within the manuscript and its Supporting information files.

**Funding:** The author(s) received no specific funding for this work.

**Competing interests:** The authors have declared that no competing interests exist.

to esophageal epithelial remodeling, hinting at metaplasia-like processes in chronic EoE. In contrast, EoE-like esophagitis was primarily immune-driven, marked by *CXCR3* ligand activation (*CXCL9, CXCL10, CXCL11*) and immunoglobulin complex enrichment, indicating systemic immune dysregulation and a potential precursor state to conventional EoE. Lymphocytic esophagitis demonstrated unique signature of metabolic dysfunction, with downregulation of oxidative phosphorylation genes (*NDUFB2, ATP5F1B*) and enrichment of neuro-immune signaling pathways, pointing at interplay between mitochondrial impairment and esophageal sensory dysfunction. A robust interferon-mediated immune response (*STAT1, IRF1* and *CXCL10*) further differentiated the latter subtype from Th2-driven conventional EoE. Nonspecific esophagitis exhibited a dominant humoral immune response, enriched for immunoglobulin-related pathways, suggestive of a B-cell-driven inflammatory mechanism distinct from the T-cell-dominated responses of other EoE subtypes.

## Conclusions

This study unveils novel molecular and immunological distinctions between conventional EoE and its subtypes. We propose that EoE-like esophagitis represents an early immune-activated phase, while lymphocytic and nonspecific esophagitis exhibit distinct metabolic and humoral immune dysregulations, respectively. Key biomarkers, *POSTN, DNAH11, CDX2*, and zinc transporters, offer critical insights for improving diagnostic criteria and guiding therapeutic approaches. These findings highlight the importance of subtype-specific therapeutic interventions and warrant longitudinal studies to map disease trajectories and therapeutic responses across EoE and its variants.

## Introduction

Eosinophilic esophagitis (EoE) is a chronic, immune-mediated disorder of the esophagus, marked by eosinophilic infiltration of the epithelium, often triggered by allergen exposure. This condition leads to dysphagia, food impaction, and esophageal remodeling [1,2]. The condition is strongly associated with atopic diseases, with over 70% of affected individuals also having asthma, eczema, or allergic rhinitis [1,2]. Once considered rare, EoE is now increasingly recognized, with rising incidence globally and significant impacts on quality of life and healthcare utilization [1]. Despite advances in clinical recognition and treatment, disease heterogeneity remains a major barrier to personalized therapy and disease classification [3]. Subtypes including EoE-like esophagitis, lymphocytic esophagitis, and nonspecific esophagitis have emerged to describe patients with overlapping clinical features but distinct histological profiles [1,3].

Although conventional EoE is well-characterized by its hallmark histological and clinical features, emerging evidence reveals its heterogeneity, encompassing a variety of atypical variants. These variants, such as lymphocytic esophagitis and nonspecific esophagitis, exhibit diverse histological, clinical, and molecular profiles [4,5].

Conventional EoE is driven by Th2-mediated inflammation, with cytokines such as interleukin-4 (IL-4), interleukin-5 (IL-5), and interleukin-13 (IL-13) aiding in recruitment of eosinophils and sustaining tissue inflammation. These cytokines disrupt epithelial barrier integrity and promote fibrosis, leading to esophageal remodeling and stricture formation [6,7]. In contrast, EoE-like esophagitis shares clinical symptoms such as dysphagia and esophageal inflammation but lacks the eosinophilic infiltration threshold for a definitive diagnosis. This entity may represent an early or milder stage of EoE or fall within a broader spectrum of eosinophilic disorders [8,9]. EoE-like esophagitis is estimated to account for 10–20% of suspected EoE cases and shares demographic and atopic associations with conventional EoE [10].

Lymphocytic esophagitis is distinguished by increased intraepithelial lymphocytes instead of eosinophils. Symptoms, including dysphagia and chest pain, overlap with those of classical EoE and gastroesophageal reflux disease (GERD). While the etiology of lymphocytic esophagitis remains unclear, it is associated with immune-mediated conditions such as celiac disease and Crohn's disease, suggesting overlapping immunopathogenic pathways [11]. Lymphocytic esophagitis is rare, with an estimated prevalence of 0.1–0.3% among individuals undergoing esophageal biopsy [12].

Nonspecific esophagitis includes cases of esophageal inflammation that do not meet the histological criteria for conventional EoE or other defined conditions. This variant often overlaps clinically with GERD, conventional EoE, and functional esophageal disorders, complicating its differentiation. Commonly associated with exposure to irritative agents like NSAIDs or alcohol, as well as systemic conditions such as autoimmune diseases, nonspecific esophagitis represents a small fraction of esophagitis cases diagnosed via biopsy and demonstrates no clear age or gender predilection [4].

Despite advances in understanding EoE pathogenesis, the molecular distinctions across its variants remain poorly defined. Conventional EoE is driven by a Th2-skewed immune response, but some atypical forms may exhibit mixed or even Th1/Th17 immune activation, indicating distinct immunological drivers [3,13]. Additionally, epithelial barrier dysfunction, including reduced expression of proteins like filaggrin and desmoglein-1 [14], contributes to esophageal inflammation and remodeling, suggesting potential molecular differences among EoE variants [15,16].

Phenotypic diversity of EoE is shaped by both genetic and epigenetic factors. Genome-wide association studies (GWAS) have identified susceptibility loci, such as *CAPN14*, a gene specifically expressed in the esophageal epithelium and regulated by IL-13 [17]. Additionally, epigenetic regulation of inflammatory and epithelial repair pathways highlights the complexity of EoE pathogenesis and its variants [18]. Therapeutic approaches for EoE, including dietary elimination, proton pump inhibitors (PPIs), and corticosteroids, show variable efficacy. Tailored therapies targeting specific molecular mechanisms are needed to improve outcomes [19]. Investigating the molecular and immunological across EoE variants could uncover novel biomarkers and therapeutic targets, advancing individualized care.

Although conventional EoE, EoE-like esophagitis, lymphocytic esophagitis, and nonspecific esophagitis differ in histologic thresholds and inflammatory profiles, they share common molecular underpinnings. Immunologically, all forms exhibit features of Th2-mediated inflammation, including elevated levels of IL-4, IL-5, and IL-13, as well as infiltration of Th2 cells and innate immune effectors [20,21]. Genetically, susceptibility loci such as *CAPN14*—a gene highly expressed in the esophagus and regulated by IL-13—have been implicated across EoE phenotypes, suggesting shared heritable risk factors (Kottyan et al., 2014; Litosh et al., 2017)[22,23]. Epigenetically, studies have uncovered disease-associated DNA methylation signatures and chromatin alterations affecting epithelial barrier and immune regulation in EoE patients [18]. These shared molecular features support a unifying pathogenic framework and strengthen the rationale for comparative molecular profiling across the EoE spectrum.

This study aimed to identify and characterize the molecular and immunological distinctions between conventional eosinophilic esophagitis (EoE) and its subtypes—EoE-like esophagitis, lymphocytic esophagitis, and nonspecific esophagitis—by integrating multi-omics analyses to refine disease classification and improve diagnostic precision. Using a multi-omics approach, including gene expression analysis, Weighted Gene Co-Expression Network Analysis (WGCNA), functional enrichment studies, and machine learning, we identified distinct molecular signatures for each subtype. Conventional EoE exhibited upregulation of periostin (*POSTN*), indicating extracellular matrix remodeling, alongside

suppressed zinc-related pathways, suggesting epithelial barrier dysfunction. EoE-like esophagitis was primarily immune-driven, characterized by *CXCR3* ligand activation and immunoglobulin enrichment, indicative of systemic immune dys-regulation. Lymphocytic esophagitis displayed mitochondrial dysfunction and oxidative phosphorylation deficits, coupled with a strong interferon-mediated immune response, while nonspecific esophagitis was marked by a humoral immune response dominated by immunoglobulin-related pathways. *FOXA1* emerged as a key transcriptional regulator influencing inflammation across subtypes. These findings highlight the molecular heterogeneity of esophageal inflammatory disorders, suggesting that EoE-like esophagitis may represent an early systemic immune response preceding conventional EoE, while lymphocytic and nonspecific esophagitis exhibit distinct immune and metabolic dysregulations. This study underscores the importance of subtype-specific diagnostic and therapeutic strategies and calls for further longitudinal research to define disease progression pathways.

## Materials and methods

### Data collection

We obtained bulk RNA sequencing datasets from the Genome Expression Omnibus (GEO), containing esophageal tissue sample derived from patients with conventional EoE and its variants: EoE-like esophagitis, lymphocytic esophagitis and nonspecific esophagitis. We used dataset GSE148381 ([https://www.ncbi.nlm.nih.gov/geo/query/acc.cgi?acc=GSE148381](https://www.ncbi.nlm.nih.gov/geo/query/acc.cgi?acc=GSE148381)). GSE148381 utilized esophageal biopsy samples collected during endoscopic procedures from 10 conventional EoE patients, 13 EoE-like esophagitis patients, 5 lymphocytic esophagitis patients, 10 nonspecific esophagitis patients, 6 gastroesophageal reflux disease (GERD) patients, and 7 healthy controls. we excluded the 6 GERD patients from our analysis. The platform used for gene expression profiling was GPL24676 Illumina NovaSeq 6000 (Homo sapiens). During preprocessing, genes with zero counts across all samples were removed to ensure robust analysis. See Fig 1, which summarizes the sequence of our research methodology. After preprocessing and quality control, four analyses—differential expression, weighted gene co-expression network analysis (WGCNA), transcription factor activity inference, and machine learning classification—were applied in parallel, each independently interrogating the same dataset.

This investigation was conducted entirely in silico (dry lab), using publicly available transcriptomic datasets and computational analyses without the generation of new biological samples. All methodological steps were performed using bioinformatics pipelines and computing environments.

### Differential gene expression

Differentially expressed genes (DEGs) exhibit significant changes in expression between conditions, providing insights into biological processes, disease mechanisms, and drug responses [24,25]. We used the "DESeq2" package [25] within the R software environment identify genes displaying differential expression between patients with conventional EoE, its variants and healthy controls. All gene expression values were subjected to log2 transformation for normalization. Genes exhibiting |log2 fold change (FC)| > 1.5 and a p-adjusted value of $p < 0.05$ were considered statistically significant. All statistical significance values represent Benjamini–Hochberg–adjusted p-values ($p_{adj}$) to control for false-discovery rate across multiple comparisons. This cutoff was chosen to balance statistical confidence with biological relevance, ensuring robust gene-level changes for downstream analyses. While alternative thresholds could be applied, this choice provided a stable and interpretable gene set, consistent with prior transcriptomic studies (Mittal et al., 2025; Petrenko et al., 2024)[26,27]. Notably, genes with smaller p-adjusted values were accorded higher rankings.

### Gene ontology and functional analysis

To investigate the biological significance of key genes, we performed Gene Ontology (GO) and functional enrichment analyses using Gene Set Enrichment Analysis (GSEA), implemented via the R packages DOSE, org.Hs.e.g.,db, and enrichplot

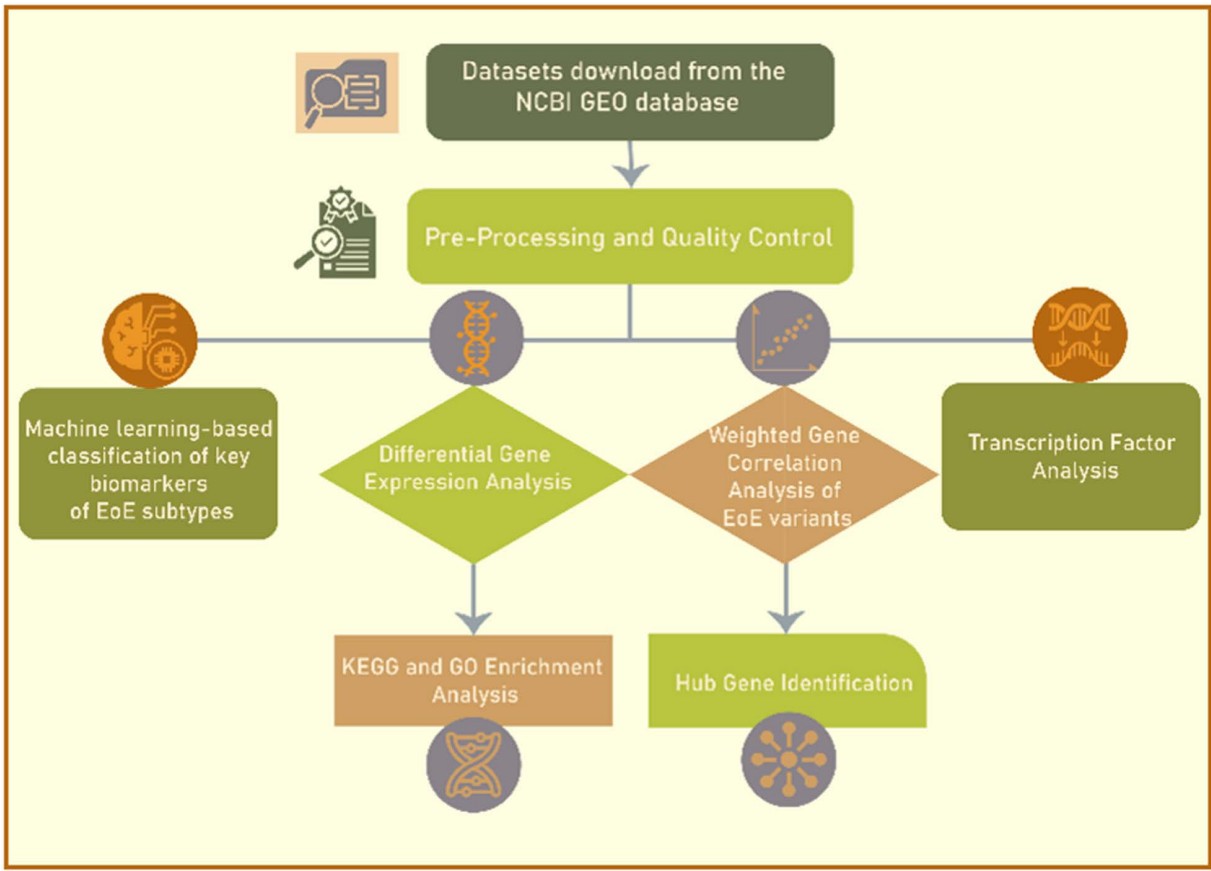

**Fig 1. Flow chart of the research methodology.** This study entailed data acquisition, preprocessing, differential gene expression analysis, weighted gene co-expression network analysis (WGCNA), gene ontology (GO) enrichment and transcription factor analysis for distinguishing molecular and immunological signatures across different subtypes of esophagitis.

[28–30]. For our study, we selected sets of differentially expressed genes as our test data, applying a significance threshold of p < 0.05 and log2 fold change (FC) > 1.5. We identified statistically significant gene enrichment pathways, covering Biological Process (BP), Cellular Component (CC) and Molecular Function (MF) in the GO enrichment analysis. The Jaccard similarity analysis was subsequently employed to compare enriched biological pathways across conventional EoE and the different eosinophilic esophagitis variants, and with the goal of identifying overlapping and distinct enriched biological GO terms across these conditions. By analyzing the set of significant pathways for each patient group, the Jaccard index quantified the proportion of shared pathways relative to the total number of pathways across groups [31]. This analysis revealed key similarities between subtypes, such as shared inflammatory and immune response pathways, while also highlighting unique processes specific to certain patient groups. Despite the lack of statistical significance, pathway overlap provided insights into shared disease mechanisms and differentiation. This approach underscores the utility of pathway similarity metrics in understanding complex relationships among related yet distinct diseases.

## Construction of weighted gene co-expression network

For Weighted Gene Co-Expression Network Analysis, the block-wise approach was employed to construct separate co-expression networks for distinct sample groups or conditions [32–34]. This strategy enables the identification of

condition-specific modules and the detection of relationships specific to each group. The block-wise WGCNA process entails data preprocessing, correlation calculation, construction of an adjacency matrix, thresholding, calculation of the Topological Overlap Measure (TOM), hierarchical clustering, and determination of module eigengenes and module membership [35]. To initiate the analysis, an expression matrix was created using Differentially Expressed Genes (DEGs), with columns representing samples or conditions and rows representing genes. Rigorous quality checks were conducted to identify and eliminate outliers or low-quality samples, ensuring the robustness of subsequent network analysis [36,37]. Subsequently, pairwise correlations between genes were computed utilizing the Pearson correlation coefficient [38].

## Selection of key modules based on clinical traits

Key modules were selected based on clinical traits using WGCNA. Given the large-scale nature of the gene expression dataset, the block-wise approach was employed to enhance computational efficiency and handle memory constraints. This method allows for the construction of networks in smaller, manageable blocks while preserving the accuracy of co-expression relationships [32]. The module eigengenes were correlated with the clinical traits of interest. This was done by calculating the correlation coefficient between the module eigengene values and the corresponding clinical trait values. Positive correlations indicate a positive association between the module and the clinical trait, while negative correlations indicate a negative association. Statistical tests determine significant module-trait associations and commonly used methods include linear regression analysis or permutation tests [34]. Key modules were selected based on their module significance and gene significance. Modules with high module significance and genes with high gene significance values are considered important for the clinical traits of interest. These key modules represent gene expression patterns that are strongly associated with the clinical phenotype [39].

## Identification of transcription factors (TFs) within the key modules

The identification of transcription factors (TFs) within key modules was performed using an R-based computational framework, leveraging curated TF-target interaction networks and transcriptomic analysis. Differentially expressed genes (DEGs) were identified using DESeq2 [25], an established statistical method for RNA-Seq analysis that estimates fold change and dispersion across experimental conditions. To infer TF activity, we employed decoupleR [40], which integrates multiple computational strategies to estimate regulatory influence. Specifically, we utilized the Dorothea database [41], a curated collection of TF-target interactions, in conjunction with weighted mean (wmean) and viper (Virtual Inference of Protein-activity by Enriched Regulon analysis) methods to assess transcriptional regulatory patterns. This approach allowed us to infer which transcription factors were most likely driving the observed gene expression changes. The resulting TF activity scores were then visualized through hierarchical clustering and heatmaps using pheatmap [42], enabling the identification of distinct regulatory patterns across conditions. This data-driven approach ensures that TF identification is based on quantitative inference from experimental RNA-Seq datasets, rather than pre-built network models, providing robust insights into key transcriptional regulators in disease-associated modules.

## Machine learning-based classification of EoE and variants

For this methodology, a total of 51 patient samples were analyzed, each characterized by 36,867 gene expression levels. The samples represented various conditions, including conventional esophagitis, EoE-like, lymphocytic and non-specific esophagitis, as well as healthy controls. To identify distinct patterns within the dataset, Principal Component Analysis (PCA) was performed to reduce dimensionality while retaining the maximum variance in gene expression data [43]. Following dimensionality reduction, the optimal number of clusters was determined using the Elbow Method and Silhouette Analysis, which suggested three clusters as the best representation of the dataset (see S3A–B Fig). The Elbow Method involves plotting the within-cluster sum of squares (WCSS) against the number of clusters and identifying the point where

adding more clusters results in diminishing returns [43,44]. Additionally, Silhouette Analysis was used to measure how well-separated the clusters were, with higher silhouette scores indicating more distinct clustering structures [45]. K-means clustering was then applied to naturally discover the groupings within the samples based on their gene expression profiles [46]. For supervised classification, a Random Forest classifier was utilized to evaluate the importance of genes in distinguishing between some selected patient groups based on the clustering patterns [47]. The model was evaluated under a 5-fold stratified cross-validation framework to ensure balanced representation of both groups across folds. Performance metrics, including accuracy and area under the ROC curve (AUC), were computed for each fold and summarized as averages with standard deviations.

The classifier was further employed to rank genes by their contribution to classification accuracy, and the top 10 biomarkers associated with the various conditions were extracted for subsequent biological pathway analysis.

### Construction of protein-protein interaction and identification of hub genes

To construct the PPI of the modules of interest, we first inputted the gene lists from the WGCNA modules of interest into the STRING (https://string-db.org/) plugin in CytoScape to construct as protein-protein interaction (PPI) network. The PPI network was analyzed and visualized by Cytoscape software (http://www.cytoscape.org/). Within Cytoscape, we harnessed the cytoHubba plugin to identify the hub genes of the modules of interest. For our specific analysis, we applied the Degree Centrality (DC) algorithm to rank the nodes (genes) according to their significance as hub genes within the PPI network [48]. DC was chosen over other hub detection methods due to its computational efficiency, ability to highlight genes with extensive direct interactions, intuitive interpretation, and robustness in identifying biologically significant regulatory genes within large PPI networks, making it particularly well-suited for our analysis [49,50].

## Results

### Identification of DEGs

DEG analysis was performed on a total of 45 esophageal biopsy samples, comprising 10 conventional EoE, 13 EoE-like esophagitis, 5 lymphocytic esophagitis, 10 nonspecific esophagitis, and 7 healthy control samples derived from the GSE148381 dataset. Using an adjusted p-value < 0.05, all patient groups showed differential gene expression relative to healthy controls. This confirms that each group diverges from the control state. The top 10 DEGs for the various patient groups are shown in the S1A–D Fig) and a summary of the DEG patterns and top 50 DEGs between the patient groups and healthy controls are also shown in Fig 2A–E. These broad transcriptional shifts confirm that each esophagitis subtype has a distinct molecular profile. The profiles include shared and unique patterns, likely reflecting different underlying mechanisms.

### Functional enrichment analysis of DEGs

The KEGG pathway was analyzed to explore the biological function of upregulated genes and downregulated genes in DEGs. As shown in Fig 2A–D, our analysis revealed lymphocytic esophagitis to be significantly associated with energy dysregulation, as evidenced by the significant enrichment of biological processes related to oxidative phosphorylation (GO:0006119), proton motive force-driven ATP synthesis (GO:0015986), and cytoplasmic translation (GO:0002181). Suppression of these pathways suggests mitochondrial dysfunction and impaired energy metabolism in lymphocytic esophagitis. These deficits could affect esophageal sensory and contractile function. The downregulation of these pathways is driven by key genes such as *NDUFB2, NDUFS3, COX7A2, ATP5F1B, ATP5PB, and NDUFA2*, which play essential roles in the electron transport chain and ATP synthesis. We also observed that there was enrichment of synapse assembly (GO:0007416), synapse organization (GO:0050808), neuron projection morphogenesis (GO:0048812), neuron development (GO:0048666), and neuron projection development (GO:0031175) suggesting changes in neuronal structure and

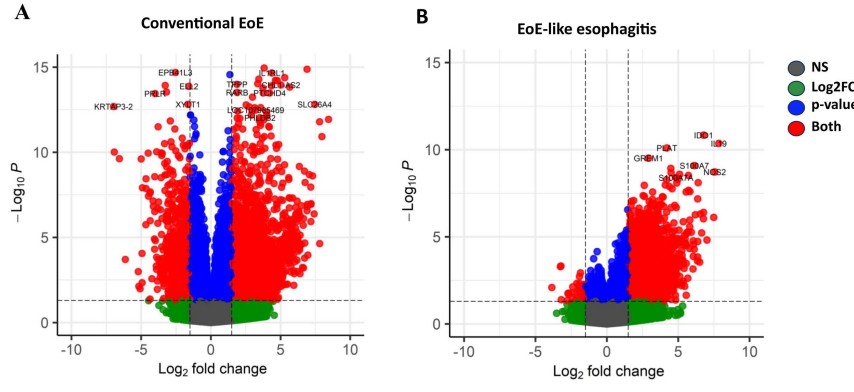

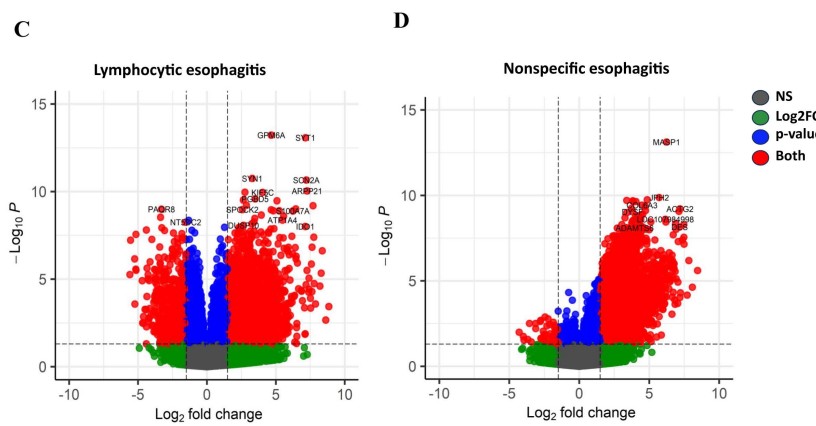

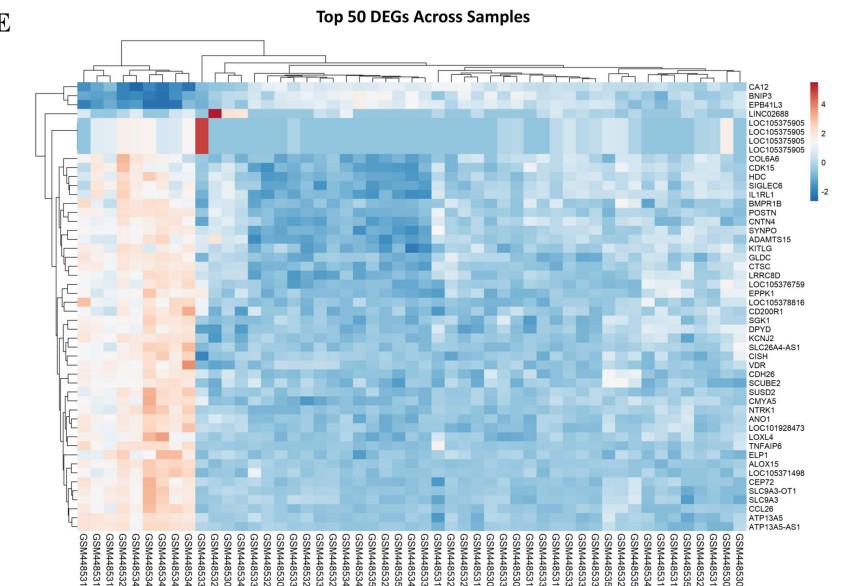

**Fig 2. Volcano plots illustrating differential gene expression between healthy controls and patients with various esophageal conditions.** (A) Conventional EoE (ce) shows a high number of significantly upregulated and downregulated genes with genes such as **ALOX15**, **LOC105375905** and **POSTN** exhibiting relatively large fold changes. (B) EoE-like esophagitis (el) presents moderate differential expression

patterns with genes such as **LOC105375905** and **SLC6A19**. (C) Lymphocytic esophagitis (ly) shows fewer significant changes, with some notable upregulated genes such as **LOC105375905**, **TRIM15** and **ZIC4**. (D) Nonspecific esophagitis (ns) displays distinct gene expression patterns with significant upregulation and downregulation of **LOC105375905**, **SLC6A19** and **C11orf86**. Red dots represent genes that meet significance thresholds for both p-value and fold change, while blue and green represent genes meeting one of the thresholds. Note: Apparent differences in the contour ("belt shape") of the plots reflect the inherent distribution of differential expression within each patient group, rather than methodological variation. (E) Heatmap showing the expression profiles of the top 50 differentially expressed genes (DEGs) across samples. Rows represent individual genes, while columns correspond to patient samples from various conditions. The color gradient indicates expression levels, with red denoting upregulation and blue denoting downregulation.

connectivity in the esophageal tissue. These processes are driven by genes including *GAP43, ASIC2, LRRTM1, NRG3, SLITRK3, SYNDIG1, NTRK3, RIMS1, STMN2, SYT1, LHX2, SULT4A1, CSMD3,* and *UNC5A*, which are involved in synaptic plasticity, axonal growth, and neurodevelopment.

In addition, the analysis of the GO term similarity matrices for biological processes cellular components and molecular functions for the EoE subtypes revealed distinct clustering patterns and varying correlation levels. These clustering differences likely mirror distinct disease mechanisms, with variation in immune activation, epithelial barrier involvement, and tissue remodeling pathways shaping the molecular signatures of each subtype.

Overall, correlations between the variants were generally low, indicating limited similarity among terms. Interestingly, in terms of cellular components, nonspecific esophagitis and EoE-like esophagitis had the highest correlation (0.56), clustering closely, with a strong overlap in extracellular structures, particularly within the extracellular region, extracellular space, and immunoglobulin complexes, suggesting that immune-related proteins and extracellular matrix components play a crucial role in both conditions. There is also the presence of immunoglobulin-associated genes like *IGHA1, IGHG1, IGKC*, and *IGLC1* in both conditions. Additionally, there is the presence of brush border membrane (*SLC6A19*) in EoE-like esophagitis but not in nonspecific esophagitis.

In terms of molecular function, lymphocytic esophagitis and conventional EoE had a relatively higher correlation (0.24), while the cases of EoE-like esophagitis were moderately similar to conventional EoE (0.19). ([Fig 3E]). The overlap points between lymphocytic esophagitis and conventional EoE points to partially shared regulatory and receptor-mediated processes between these conditions, despite their differing dominant immune profiles.

The comparison between lymphocytic esophagitis and conventional EoE in terms of molecular function revealed key shared and distinct regulatory mechanisms, particularly in transcription factor activity and receptor signaling. Genes such as *RORB* and *NR4A1* appeared in both conditions but were linked to different GO terms. Nonspecific esophagitis remained distinct, with near-zero correlations (see [S4A–D Fig]). Notably, lymphocytic esophagitis was enriched for DNA-binding transcription factor activity, whereas conventional EoE showed a notable presence of transmembrane signaling receptor activity and extracellular matrix structural constituents. Interestingly, we also observed presence of *CDX2* in methyl-CpG binding in conventional EoE.

The analysis of GO terms related to epithelial barrier dysfunction in EoE-like esophagitis compared to conventional EoE reveals notable differences in their cellular component profiles (see [S5 Fig]). In conventional EoE, there is a significant downregulation of the cornified envelope (GO:0001533), a key structure responsible for epithelial barrier integrity. This is accompanied by the downregulation of structural proteins related to desmosomes and adhesion molecules such as *DSP, PKP3, PPL, SCEL, IVL,* and *TGM1*, which are essential for maintaining the strength and cohesion of the epithelial layer. Additionally, there is an upregulation of cellular components associated with plasma membrane signaling, suggesting increased immune cell interactions, possibly as a response to epithelial damage. In contrast, the EoE-like esophagitis dataset does not exhibit the same degree of barrier dysfunction. There is no significant depletion in cornified envelope components, indicating that the epithelial structure remains relatively intact. However, there is an enrichment of genes associated with the extracellular region and extracellular space.

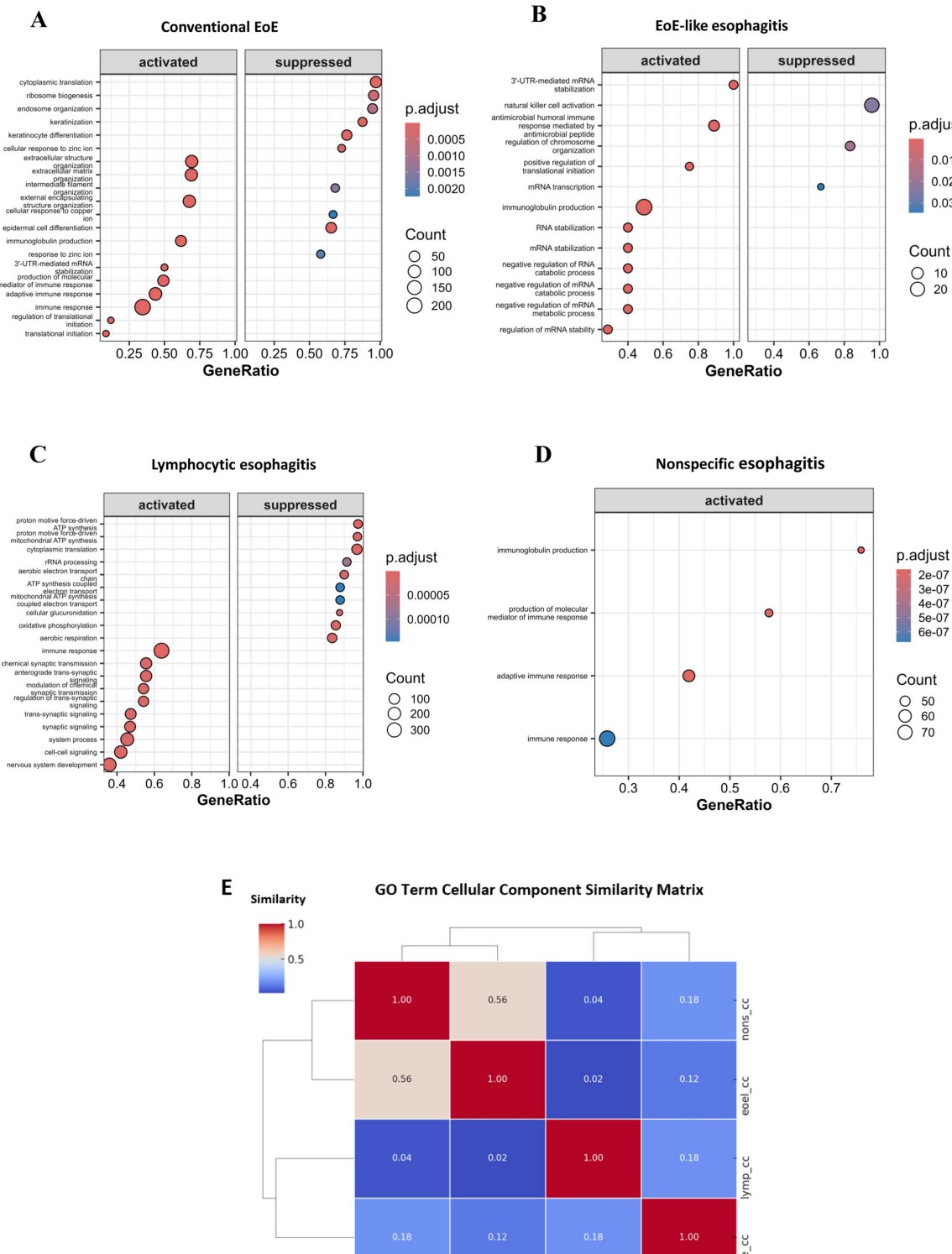

**Fig 3. Enrichment analysis of biological processes for different esophageal conditions.** The x-axis represents the gene ratio, while the dot size indicates the number of genes associated with each term, and the color reflects the adjusted p-value. (A): Biological processes in conventional EoE (CE) reveal significant activation of processes such as cytoplasmic translation, immune response, and keratinization, while suppression involves translational

initiation and regulation. (B) EoE-like esophagitis (EL) displays enrichment in translation activator activity and signaling receptor activity. (C): Lymphocytic esophagitis (LE) highlights activation in processes related to mitochondrial ATP synthesis, oxidative phosphorylation, and nervous system development, with suppression in trans-synaptic signaling. (D) Nonspecific esophagitis (NS) shows activation in immune-related processes, including immunoglobulin production and adaptive immune response. (E) GO term Cellular Component Jaccard similarity matrix. The heatmap displays the pairwise similarity of cellular component GO terms among nonspecific esophagitis (nons_cc), EoE-like esophagitis (eoel_cc), lymphocytic esophagitis (lymp_cc), and conventional EoE (ceoe_cc). The similarity scores range from 0 (blue, low similarity) to 1 (red, high similarity). The matrix reveals distinct clustering patterns, with closer relationships observed between nonspecific and EoE-like esophagitis, compared to conditions.

Interestingly, The GO analysis for nonspecific esophagitis highlighted immunoglobulin production (*IGLV2-11, IGLV4-60, IGKV1-5, IGKV6-21, IGHG1),* immune response (*IGLV3-27, IGHV3-49, CLEC4E, IGKV2-30*), and antigen binding (*IGLV3-27, IGKV1-5, IGHV3-11*) as key processes, with enrichment in immunoglobulin complexes and blood microparticles, emphasizing a strong immunoglobulin-driven extracellular immune activation.

**Weighted co-expression network construction**

We applied the WGCNA algorithm to construct a co-expression network and identify modules within the gene expression data from the 51 samples. To achieve a scale-free topology with a coefficient of determination ($R^2$) of 0.80 (Fig 4A, 4B), the Pearson's correlation matrix of the genes was converted into a strengthening adjacency matrix using a power parameter (β) of 9. The β value was selected using the scale-free topology criterion, where different soft-thresholding powers were tested, and the lowest β value that maintained approximately scale-free topology ($R^2 \geq 0.80$) was chosen. The Dynamic Tree Cut algorithm was then employed to cluster the selected genes based on a topological overlap matrix (TOM)-based dissimilarity measure, leading to the division of the gene tree into 50 distinct modules (Fig 4C). Each module was represented by a unique color and contained a specific number of genes. The Dynamic Tree Cut method was chosen over traditional hierarchical clustering approaches because it offers improved sensitivity to outliers, flexibility in detecting modules of varying sizes, and the ability to adaptively merge closely related branches [34].

Next, we analyzed the interactions among these co-expression modules using Pearson's correlation coefficient. Hierarchical clustering of module eigengenes, which summarizes the modules, revealed meta-modules represented by branches in the dendrogram (Fig 4C). These meta-modules were grouped based on the correlation of eigengenes. Finally, to visualize the topological overlap between the genes within each module, we generated a heatmap plot. The heatmap showed different gene clusters within each module, with positive correlations represented in red and negative correlations in blue. The modularization and identification of gene clusters based on their co-expression patterns provided valuable insights into significant associations between these clusters and the healthy controls (normal), eosinophilic-like esophagitis (el), nonspecific esophagitis (ns), lymphocytic esophagitis (ly), and conventional esophagitis (ce) groups. The analysis revealed 26 module-trait pairs with statistically significant correlations ($p < 0.05$).

The MElightgreen (Fig 4E) and MEwhite modules exhibited the strongest positive correlation with conventional esophagitis ($r = 0.86$, $p < 0.001$) and ($r = 0.74$, $p < 0.001$) respectively, whilst MElightsteelblue1 displayed a high positive correlation with lymphocytic esophagitis ($r = 0.51$, $p < 0.001$). In nonspecific esophagitis patients, the MEsalmon4 module showed a significant positive correlation ($r = 0.31$, $p < 0.05$) (Fig 4D).

These results suggest that specific gene modules are differentially associated with the distinct esophagitis subtypes and healthy controls, highlighting the potential for identifying unique molecular signatures across these conditions. After identifying MElightsteelblue1 and MEwhite as modules of interest, we performed GO enrichment analysis. This uncovered functionally enriched pathways associated **with** these modules. Again, our analysis revealed that the MEwhite module was highly enriched for genes involved in mitotic cell cycle, chromosomal segregation, and DNA dynamics (see S2 Fig). The GO analysis also showed that MElightsteelblue1 module is strongly associated with immune system

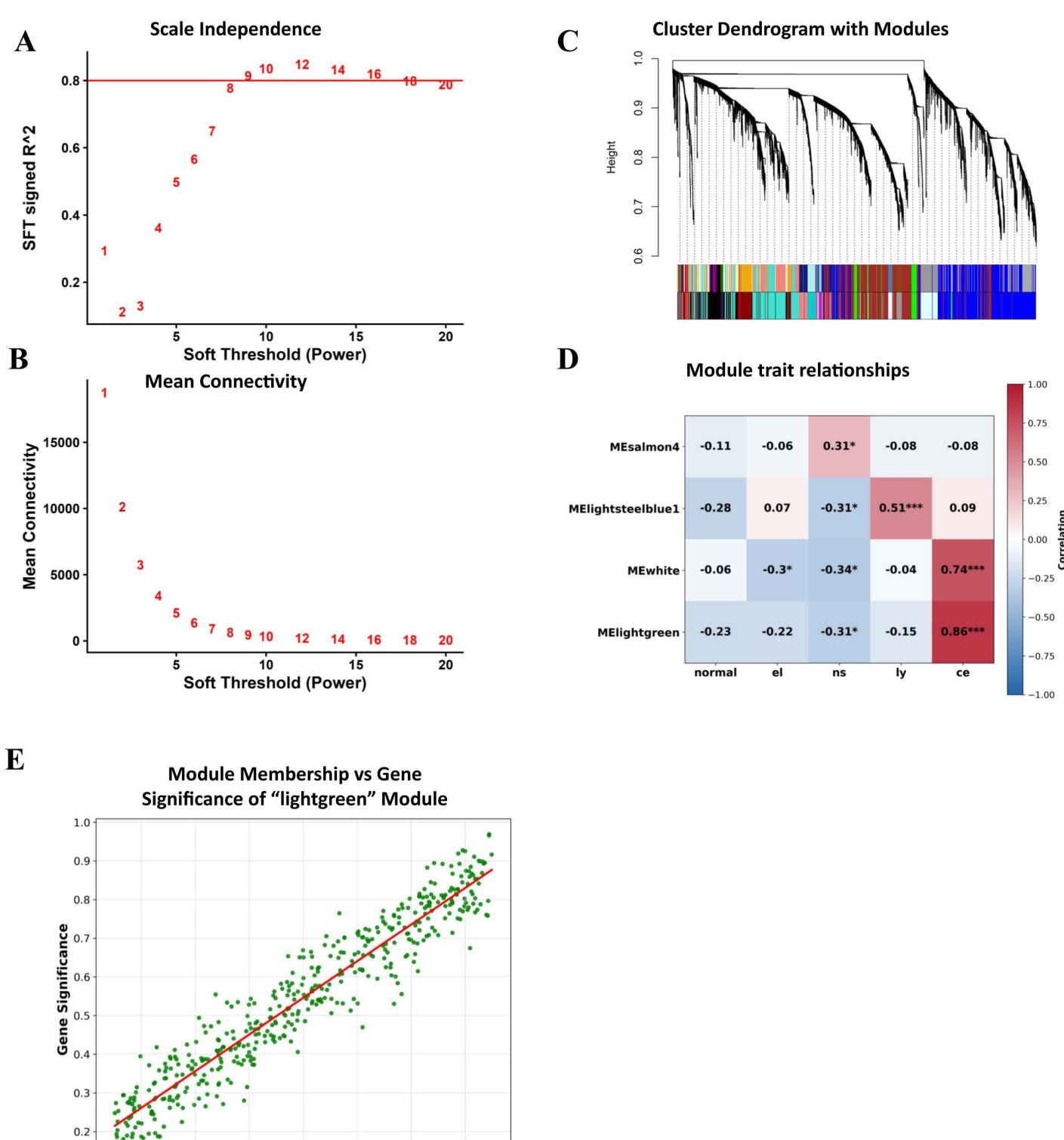

**Fig 4. Construction of co-expression modules by WGCNA.** (A, B) The adjacency matrix was defined using soft-thresholds with β = 9. Network topology analysis was performed under different soft threshold powers. (C) Clustering dendrograms of genes, with dissimilarity based on topological overlap, together with assigned module colors. (D) The Module-trait relationship heatmap illustrates the correlation between gene modules and different

disease conditions, specifically healthy controls (normal), eosinophilic-like esophagitis (el), nonspecific esophagitis (ns), lymphocytic esophagitis (ly), and conventional esophagitis (ce). Rows correspond to modules, and columns correspond to the disease conditions. Each cell contains the correlation and p-value information, with red indicating positive correlation and blue indicating negative correlation. The heatmap visualizes the module-trait correlations of key modules, with positive correlations depicted in red and negative correlations in blue. The intensity of the color reflects the strength of the correlation, with the associated p-values indicated by asterisks: *$p < 0.05$, **$p < 0.01$, and ***$p < 0.001$. (E) The correlation between gene signatures for conventional EoE and module membership in the lightgreen module.

functions, particularly those involving antigen processing, phagocytosis, and defense against viral and microbial agents, immune activation and lymphocyte infiltration.

To further integrate our DEG-GO gene enrichment with WGCNA-derived gene module lists, we compared results at the gene level. We observed partial convergence. In conventional EoE, inflammatory signaling genes such as *FOS*, *JUN*, *DUSP1*, and *CXCL8* were identified by both DEG-GO and the lightgreen WGCNA module, whereas barrier integrity genes (*IVL*, *LOR*, *SPRR1B*, *FLG*) were specific to DEG analysis. In EoE-like esophagitis, antimicrobial defense genes (*DEFB1*, *S100A8*, *S100A9*, *LCN2*) overlapped between DEG-GO and the lightsteelblue1 module, while RNA catabolism genes (*EXOSC3*, *DIS3*, *XRN1*) were unique to DEG-GO. In lymphocytic esophagitis, chemokine and interferon-related genes (*CXCL9*, *CXCL10*, *CXCL11*, *CCL5*, *STAT1*) were consistently detected by both methods, whereas mitochondrial and translation-related DEGs (*NDUFB2*, *NDUFS3*, *ATP5F1B*, *MRPL12*) were captured only by DEG-GO. In contrast, nonspecific esophagitis showed limited overlap: immunoglobulin genes (*IGHG1*, *IGKC*, *IGLC1*, *IGLV3-27*, *IGKV1-5*) were strongly enriched in DEG-GO but were dispersed across modules and did not form co-expression clusters in WGCNA.

Taken together, DEG-GO emphasizes disease-specific effector processes (e.g., epithelial barrier disruption, oxidative phosphorylation). By contrast, WGCNA-GO highlights coordinated immune signaling hubs such as chemokines.

## PPI network construction and hub gene identification

Next, we examined the protein-protein interaction (PPI) network of the identified WGCNA modules of interest using the STRING database. By integrating data from the STRING database and employing Cytoscape software, we successfully mapped 647 and 1154 differentially expressed genes (DEGs) from the MElightgreen and MElightsteelblue1 modules respectively onto the PPI network. The resulting PPI network, consisting of DEGs, identified several functionally enriched clusters ranked based on their scores. To identify hub genes for the MElightgreen and MElightsteelblue1 modules, we utilized the Degree Centrality method in CytoHubba and drilled down to the top 10 genes (ranked by the highest degree in the network). Notably the MElightgreen module was enriched with genes involved in inflammatory response (*IL13, IL5*), neurotrophic signaling (*BDNF, NTRK1*), and TGF-β signaling (*SMAD3*), suggesting its potential role in immune regulation and tissue remodeling (Fig 5A). In contrast, the MElightsteelblue1 module featured key regulators of immune activation and antiviral defense, including *STAT1, CXCL10*, and *IRF1*, which are known to drive interferon signaling and cytokine-mediated responses (Fig 5B). These interferon-inducible genes suggest that lymphocytic esophagitis is characterized by a strong interferon-mediated immune response, consistent with autoimmune-like pathology. Lastly, the MEwhite module was dominated by genes such as *CDK1, CCNB1*, and *CENPA*, which play essential roles in cell cycle regulation and chromosomal dynamics, hinting at its association with cell proliferation and mitotic progression (Fig 5C). These findings suggest that each module is functionally distinct, with MElightgreen contributing to immune and tissue remodeling processes, the MElightsteelblue1 being associated with immune signaling and antiviral responses, and the MEwhite driving cell cycle progression. These findings reinforce the role of immune signaling and tissue remodeling in conventional EoE. They also provide new insight into how conventional and EoE-like subtypes differ, particularly in cell-proliferation programs. Furthermore, these observations demonstrate that lymphocytic esophagitis is primarily driven by interferon and cytokine-mediated immune signaling, a characterization that is previously unreported in literature.

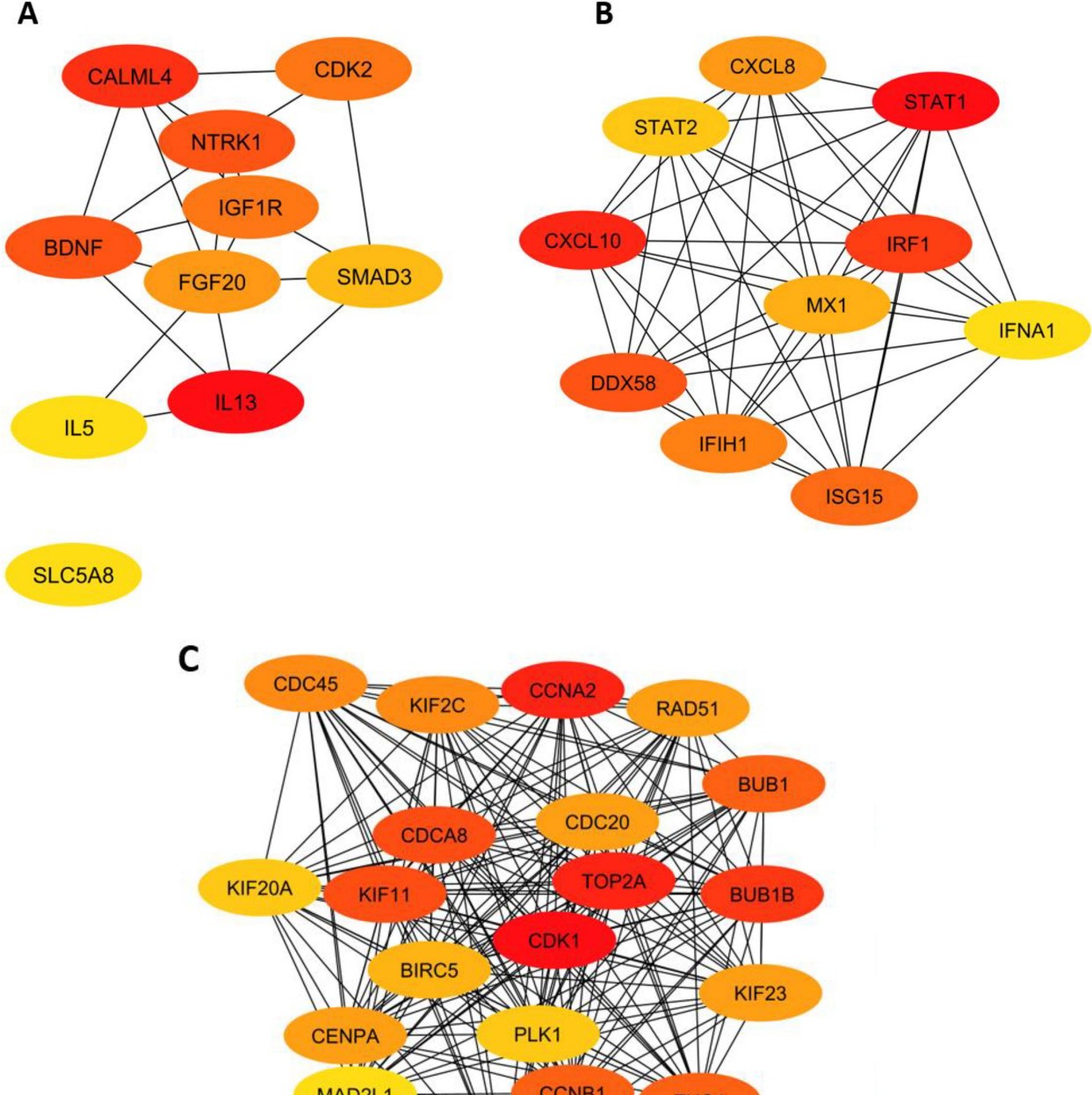

**Fig 5. Hub gene identification from selected gene modules using CytoScape plugin CytoHubba degree centrality algorithm and filtering to top 10 by degree.** (A): Hub gene signature for the MElightgreen module, shows key genes such as **IL13, NTRK1, BDNF** and **IGF1R** play a significant role in the pathophysiology of conventional EoE. (B): Hub gene signature for the MElightsteelblue1 module, shows important genes like **STAT1, CXCL10, CXCL8** play major roles in the pathophysiology of lymphocytic esophagitis. (C) Hub gene signature for MEwhite module includes key genes such as **CDK1, CCNB1, CENPA, RAD51** and **CDC20** which are involved in cell proliferation epithelial turnover. This module shows positive correlation with conventional EoE and a negative correlation with EoE-like esophagitis.

To assess the relationship between WGCNA-derived hub genes and differential expression, we cross-referenced the hub genes identified within the PPI networks of WGCNA modules against DE analysis. We found substantial convergence: in conventional EoE, hub genes from the lightgreen module (e.g., *FOS, JUN, CXCL8*) were also significantly upregulated DEGs, reinforcing their central role in inflammatory signaling. In EoE-like esophagitis, hub genes such as *S100A8, S100A9*, and *LCN2* overlapped with DEGs, highlighting shared antimicrobial defense processes. In lymphocytic esophagitis, key interferon-inducible hub genes (*CXCL9, CXCL10, STAT1)* were also robust DEGs, supporting their role as drivers of Th1-skewed immunity. By contrast, in nonspecific esophagitis, many immunoglobulin DEGs (e.g., *IGHG1, IGKC, IGLC1*) did not emerge as WGCNA hubs, suggesting that antibody gene overexpression represents diffuse activity rather than tightly co-regulated network hubs.

## ML identification of key biomarkers of EoE variants

Traditional analytical approaches have provided valuable insights **into** EoE pathophysiology. However, they often miss intricate patterns and interactions in high-dimensional data. In this research step, we leveraged machine learning methodologies to systematically identify key biomarkers that differentiate conventional EoE and its variants. A total of 51 patient samples were analyzed for this step, each characterized by 36,867 gene expression levels. These samples included conventional eosinophilic esophagitis, EoE-like esophagitis, lymphocytic esophagitis, nonspecific esophagitis, and healthy controls. To explore inherent structure within the dataset, Principal Component Analysis (PCA) was performed first, reducing dimensionality while preserving the maximum variance in gene expression data (Fig 6A). Subsequently, K-means clustering was applied to stratify the dataset into three distinct clusters, capturing the heterogeneity of gene expression across conditions (Fig 6A).

Based on the previously performed Jaccard similarity and hierarchical clustering analyses, we focused the supervised machine learning (ML) classification on conventional EoE and lymphocytic esophagitis. Data pre-processing involved encoding group labels using a label encoder, extracting gene expression values, and then filtering the dataset to include only conventional EoE and lymphocytic esophagitis cases. A Random Forest classifier was employed for classification, given its robustness in handling high-dimensional data and its ability to provide feature importance rankings. Analysis of feature importance revealed ten key biomarkers distinguishing conventional EoE from lymphocytic esophagitis. Among the top-ranked biomarkers, *LOC105369634 and PRR15L* exhibited the highest importance scores (0.02), followed by *POSTN, DNAH11*, and *NFE2*, which achieved perfect classification performance (AUC = 1.00). These genes are highly discriminative between the two subtypes, underscoring their potential as robust molecular biomarkers for differential diagnosis. The relative importance scores indicated that *LOC105369634* and *PRR15L* exhibited the highest predictive relevance (importance score = 0.02), followed by the remaining eight genes with scores of 0.01. To assess model performance, receiver operating characteristic (ROC) curve analysis was conducted for each gene. Several genes demonstrated high discriminative power, with *POSTN, PRR15L, DNAH11*, and *NFE2* achieving perfect classification performance for area under the curve (AUC = 1.00) (Fig 6B–E). The Random Forest classifier also achieved an average accuracy of 0.80 ± 0.16 and a mean AUC of 0.85 ± 0.20 across 5 stratified folds. Other notable biomarkers included *LOC105369634* (AUC = 0.97), *CHAF1A* (AUC = 0.94), *DNAH7* (AUC = 0.84), and *EPHA10* (AUC = 0.83), further confirming their relevance in distinguishing conventional EoE from lymphocytic esophagitis. *SCNM1* exhibited moderate classification performance (AUC = 0.70). The fact that *DNAH11* and *LOC105369634* were also identified in the MElightgreen module strengthens their relevance.

## Identification of transcription factor activity

Inferring transcription factor (TF) activity is critical for understanding regulatory mechanisms. This approach can also reveal potential therapeutic targets and pathways in disease progression. The heatmap displays transcription factor activity across conventional EoE (ce), EoE-like (el), lymphocytic (ly), nonspecific esophagitis (ns), and normal samples

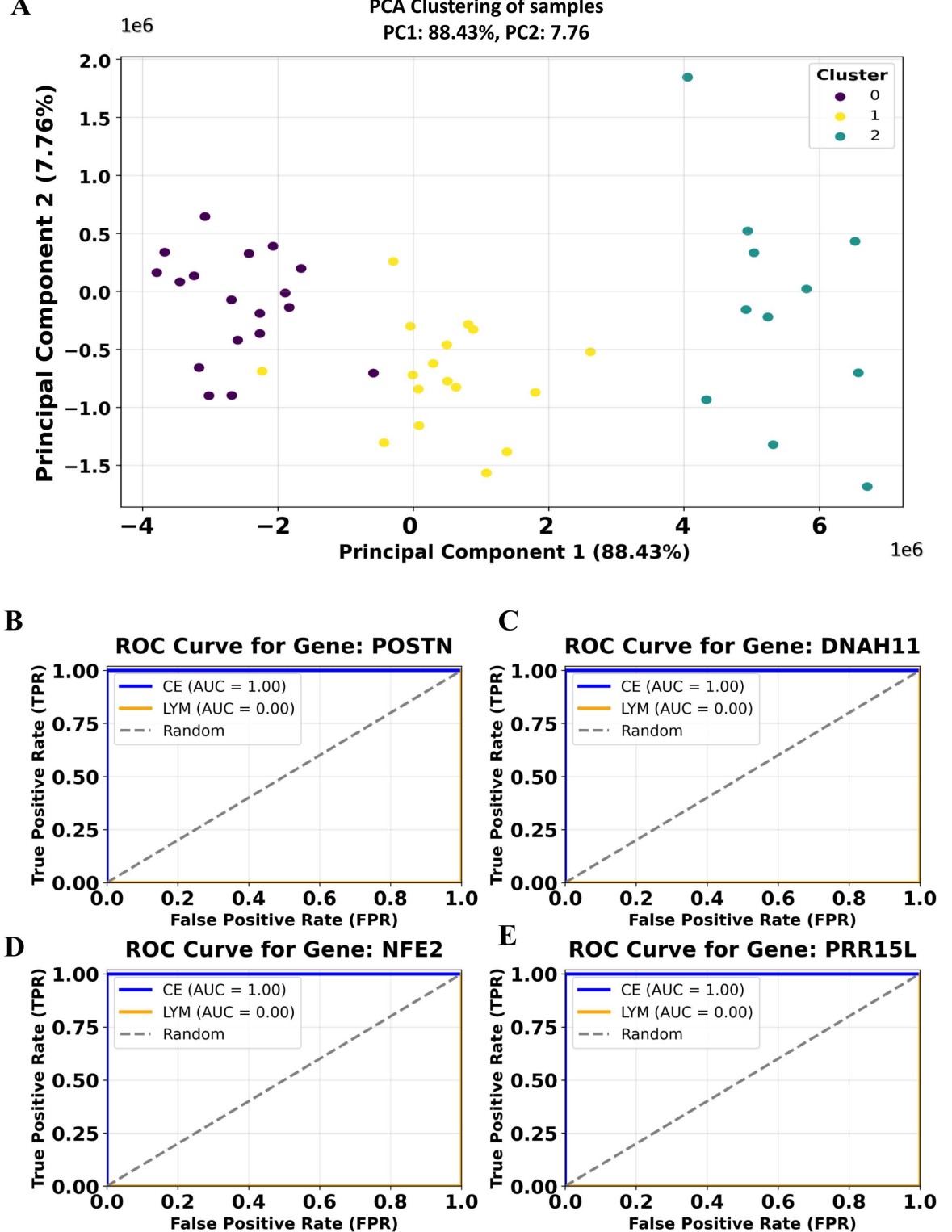

**Fig 6. ML identification of key biomarkers in EoE and its variants.** (A) Principal Component Analysis (PCA) clustering of samples into 3 groups based on gene expression data. The first principal component (PC1) explains 88.43% of the variance, while the second principal component (PC2) accounts for 7.76%. (B-E): Machine learning-based identification of key biomarkers associated with conventional EoE (CE) and lymphocytic esophagitis

(Lym) using Random Forest classification. Receiver Operating Characteristic (ROC) curves are shown for selected genes: (B) **POSTN**, (C) **DNAH11**, (D) **NFE2**, and (E) **PRR15L**. The Area Under the Receiver Operating Characteristic (AUROC) is 1.00 for CE, indicating perfect discrimination for this condition, and 0.00 for LYM, suggesting no discriminatory power for these genes in lymphocytic esophagitis. The dashed line represents random classification (AUROC = 0.5).

(Fig 7A). In conventional EoE, *FOXA1* showed the highest activity across samples, followed by *CREB1* and *STAT3*. Clustering was consistent across individuals (Fig 7B). This suggests *FOXA1* may act as a master regulator of epithelial differentiation and immune modulation in conventional EoE.

In the patients with lymphocytic esophagitis, *FOXA1* also showed high activity across all samples, with moderate activity noted for *CREB1*, *STAT3*, and *STAT1*, although slight variability in transcription factor activity was observed within the group. Some of these TFs also appeared as differentially expressed genes, including *STAT3* in EoE-like and lymphocytic esophagitis and *CREB1* in lymphocytic esophagitis. In contrast, *FOXA1, SMAD3*, and *TFAP2C* showed high inferred activity without differential expression. This suggests regulation at post-transcriptional or upstream signaling levels. A comprehensive summary of all results from the TF analysis is provided in S6 Fig.

## Discussion

Eosinophilic esophagitis (EoE) and its related subtypes—EoE-like esophagitis, lymphocytic esophagitis, and nonspecific esophagitis—represent a spectrum of immune-mediated esophageal disorders with distinct molecular and immunological profiles. This study integrates multi-omics analyses, including gene expression profiling, functional enrichment, and weighted gene co-expression network analysis to delineate the molecular underpinnings of these conditions. Our findings shed new light on molecular and immunological pathways associated with conventional EoE and its subtypes, highlighting biological similarities and distinctions among these conditions while providing new insights for diagnosis in clinical settings.

### Molecular and immunological landscape *of* conventional EoE

Our analysis confirms that conventional EoE is characterized by a Th2-driven immune response, with significant upregulation of *IL-13*, *IL-5*, and *SMAD3*, consistent with prior studies [51–53]. This inflammatory cascade promotes eosinophil recruitment, epithelial barrier dysfunction, and tissue remodeling, as evidenced by the strong upregulation of periostin (*POSTN*), a key marker of extracellular matrix deposition and fibrosis. Interestingly, our study identifies POSTN as a key biomarker of conventional EoE by way of its unique upregulation in these patients. This consistent upregulation of *POSTN* (Fig 8A) aligns with prior studies suggesting that extracellular matrix (ECM) remodeling is a major contributor to the fibrosis and structural changes observed in EoE patients [54].

Remarkably, this study also identified *DNAH11*, a gene involved in cell motility and organ morphogenesis, as a biomarker of importance in conventional EoE (Fig 8B). *DNAH11* plays a key role in epithelial turnover and tissue remodeling, potentially influencing esophageal dysmotility, which is a recognized feature of EoE [55]. This study highlights previously unreported role of *DNAH11* in maintaining barrier integrity and in remodeling of extracellular matrix in classic EoE patients. These findings suggest that, with additional validation, *DNAH11* could be developed into a biomarker for conventional EoE, particularly in identifying patients with epithelial dysfunction and esophageal dysmotility.

Importantly, in conventional EoE, Th2 cytokines (notably IL-13 and IL-5, with downstream SMAD3 signaling) connect inflammation to tissue remodeling. These signals drive eosinophil recruitment, epithelial injury, and activation of profibrotic programs, exemplified by periostin upregulation and ECM deposition. Enrichment of motility- and morphogenesis-related genes such as *DNAH11* further suggests that inflammatory cues modulate epithelial turnover and cytoskeletal dynamics, contributing to barrier dysfunction, fibrosis, and structural remodeling.

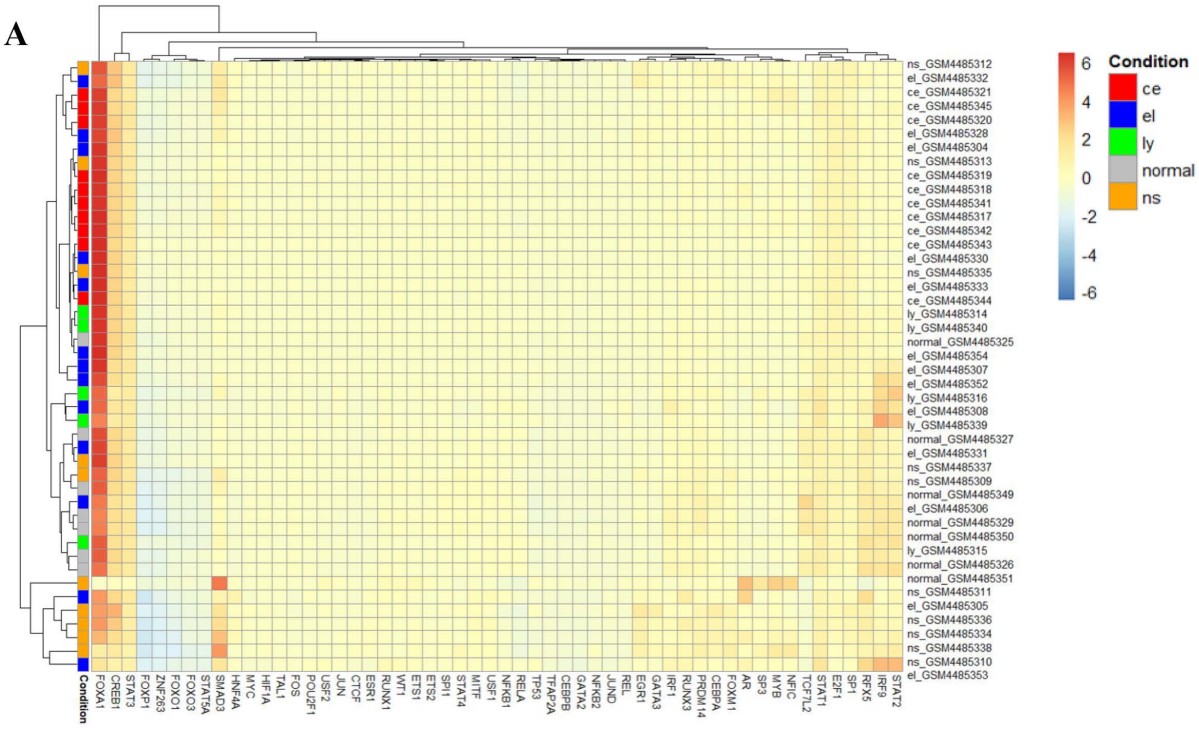

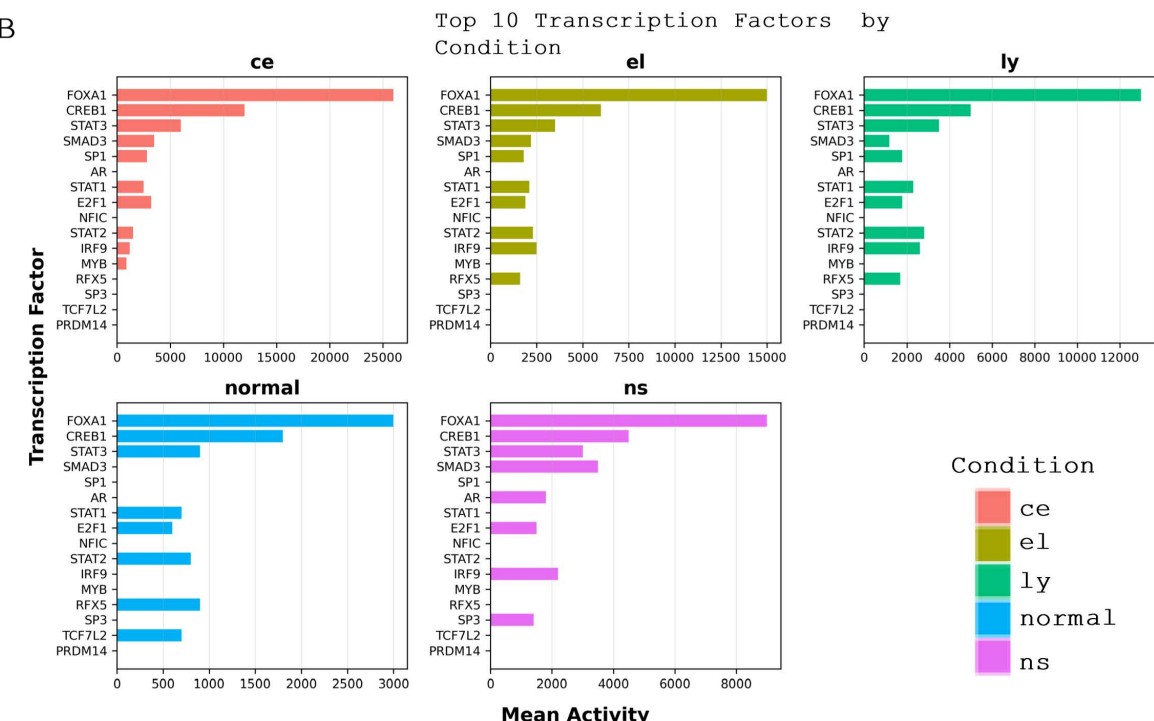

**Fig 7. Transcription factor activity in EoE and its variants.** Heatmap displaying transcription factor activity across samples. Rows represent individual samples grouped by condition, and columns represent transcription factors. Conditions are annotated on the left with color coding: red for conventional EoE (ce), blue for EoE-like esophagitis (el), green for lymphocytic esophagitis (ly), orange for nonspecific esophagitis (ns), and gray for healthy

controls (normal). The color gradient in the heatmap reflects transcription factor activity levels, with red indicating higher activity and blue indicating lower activity. Clustering reveals distinct patterns of transcription factor activity associated with specific esophageal conditions. (B) Top 10 transcription factors by condition. The bar plots display the mean activity of the top 10 transcription factors across: conventional EoE (ce, red), EoE-like esophagitis (el, olive), lymphocytic esophagitis (ly, green), nonspecific esophagitis (ns, purple), and healthy controls (normal, blue). Each plot highlights transcription factors such as **FOXA1**, **CREB1**, and **STAT3**, which demonstrate variable activity across conditions.

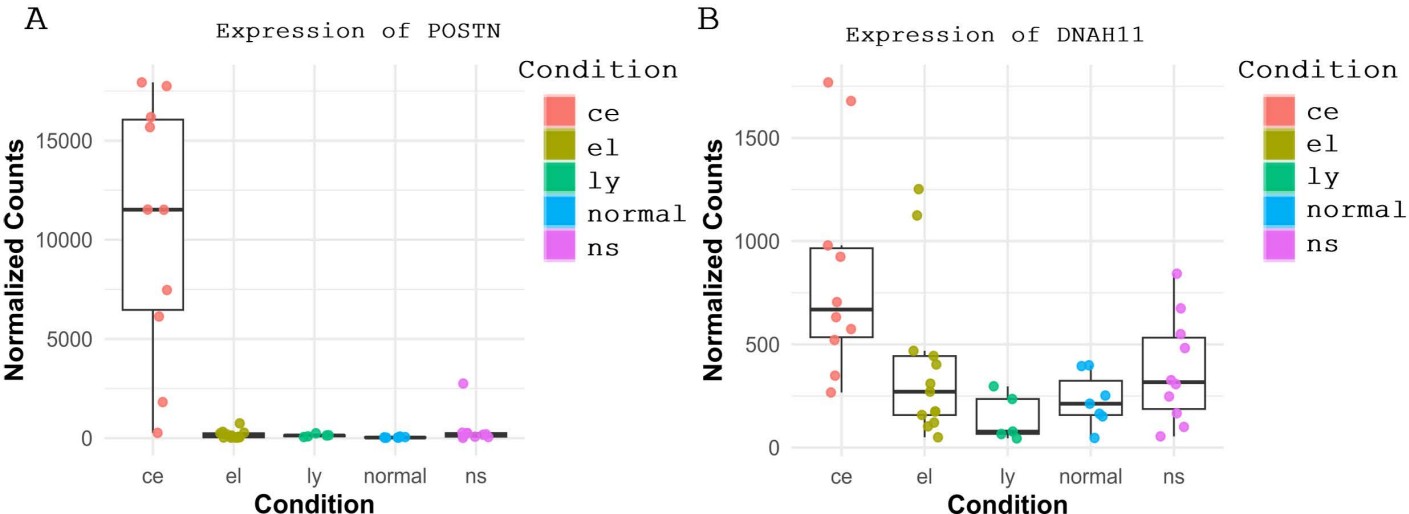

**Fig 8. Expression of POSTN (A) and DNAH11 (B) across esophageal conditions.** Both genes exhibit significantly higher expression in conventional EoE (ce) compared to EoE-like (el), lymphocytic (ly), nonspecific esophagitis (ns), and healthy controls (normal). **POSTN** demonstrates markedly elevated levels, while **DNAH11** shows more moderate expression differences across the conditions.

Additionally, the suppression of metallothioneins (*MT1X, MT1F, MT1E, MT1A, MT2A, MT1M, MT1G,* and *MT1H,*) and zinc-related pathways suggests epithelial barrier impairment, aligning with studies linking zinc homeostasis to mucosal integrity [56,57]. Metallothioneins play a crucial role in zinc homeostasis by binding and regulating intracellular zinc levels, as well as providing protection against oxidative stress [58]. Their suppression in EoE suggests a potential disruption in intracellular zinc buffering, which may contribute to epithelial barrier dysfunction and inflammatory responses commonly observed in EoE patients [59]. Further supporting the role of altered zinc homeostasis in conventional EoE, we identified differential regulation of zinc transporters. *SLC39A1* (*ZIP1*) was notably upregulated, indicating a potential compensatory mechanism to enhance zinc import under basal conditions. In contrast, *SLC39A2* (*ZIP2*) was suppressed, suggesting a selective impairment of certain zinc transport pathways. This differential regulation of ZIP transporters aligns with previous research demonstrating their role in maintaining cellular zinc homeostasis under inflammatory and pathological conditions [56]. The upregulation of *ZIP1* may be a compensatory response to maintain zinc influx, whereas *ZIP2* suppression could indicate a shift in zinc distribution or availability in esophageal epithelial cells. This finding is particularly significant in light of emerging evidence suggesting that zinc status may influence EoE severity [57].

Emerging evidence indicates that zinc status modulates T-helper cell polarization, with zinc deficiency preferentially impairing Th1 cytokine production and biasing immune balance toward Th2 inflammation [60,61]. Mechanistic studies further demonstrate that Th2/eosinophilic inflammation disrupts epithelial integrity in EoE through reduced claudin-7, suppressed HIF-1α, and dysregulated CD73/adenosine signaling [62–64]. This relationship is particularly relevant given recent evidence that zinc supplementation may reduce EoE severity [57], suggesting that modulation of zinc homeostasis

could represent a potential therapeutic strategy. While our transcriptomic results support a possible mechanistic link between zinc dysregulation and epithelial barrier impairment, further experimental validation will be needed to establish causality or therapeutic relevance.

In conventional EoE, the presence of *CDX2* expression and its association with methyl-CpG binding suggest a potential role for *CDX2*-encoded molecules in epigenetic regulation, which may contribute to disease susceptibility and persistence. *CDX2*, a transcription factor primarily involved in intestinal epithelial differentiation, is not typically expressed in the esophagus under normal conditions. Its aberrant expression in EoE raises questions about its potential role in esophageal epithelial remodeling and disease progression.

Epigenetic modifications, including DNA methylation, play a significant role in modulating gene expression, particularly in chronic inflammatory conditions such as EoE. Methyl-CpG binding proteins, such as MeCP2 and MBD2, recognize methylated CpG sites and can repress or activate gene transcription, depending on the context [65]. The identification of *CDX2* in a methylation-associated context suggests that epigenetic dysregulation may contribute to the altered differentiation of esophageal epithelial cells, potentially driving a metaplasia-like transformation that mimics aspects of intestinal differentiation. This phenomenon has been described in Barrett's esophagus, where chronic inflammation leads to the replacement of esophageal squamous epithelium with an intestinal-like phenotype, partly mediated by *CDX2* upregulation [66,67].

While EoE is not typically classified as a metaplastic disorder, the aberrant presence of *CDX2* suggests that some degree of epithelial plasticity and trans-differentiation may occur in response to chronic Th2-mediated inflammation. This aligns with prior research indicating that persistent inflammatory cytokine exposure can induce epigenetic alterations, including DNA methylation changes that affect key transcription factors regulating epithelial integrity [68,69]. Notably, IL-13, a major driver of EoE pathogenesis, has been shown to alter epithelial gene expression and may contribute to *CDX2* activation in EoE patients [52,68,69].

Another key consideration is whether *CDX2* expression in EoE is a marker of disease chronicity and treatment resistance. Given that methyl-CpG binding proteins often modulate chromatin accessibility, persistent DNA methylation changes in EoE could reinforce a pathological transcriptional pathway that sustains inflammation and epithelial dysfunction, even in the absence of continued allergen exposure [70]. This could explain why some EoE patients remain refractory to corticosteroid or dietary therapy despite a reduction in eosinophilic inflammation. Future research should focus on understanding the role of *CDX2* in EoE pathogenesis, particularly its potential as a biomarker of disease severity, persistence, and treatment resistance. Longitudinal studies tracking *CDX2* expression over time in EoE patients could determine whether its presence correlates with progression to severe or refractory disease. Additionally, identifying differentially methylated regions (DMRs) regulating *CDX2* expression and functional studies of esophageal organoids may shed some light on its role in epigenetic modifications that sustain inflammation and epithelial remodeling. Furthermore, comparative analyses with Barrett's esophagus could help clarify whether similar epigenetic mechanisms drive *CDX2* expression in both conditions or if distinct pathways underlie its role in EoE pathogenesis. These investigations will be critical for refining disease classification, identifying novel therapeutic targets, and developing precision medicine approaches tailored to particular subsets of EoE patients with distinct epigenetic signatures.

### EoE-like esophagitis: A systemic immune-driven variant

EoE-like esophagitis exhibits a predominantly immune-driven pathophysiology, with *CXCR3* ligand activation (*CXCL9, CXCL10, CXCL11*) and immunoglobulin complex enrichment, suggesting a systemic immune dysregulation rather than localized eosinophilic infiltration. These findings reinforce the idea that EoE-like esophagitis may represent an early immune-activated state that could progress to conventional EoE [1]. The presence of blood microparticles further indicates systemic immune activation, highlighting potential autoimmune-like features. Longitudinal studies indicate that a subset of patients with EoE-like esophagitis may progress to conventional EoE over time, suggesting a continuum of disease

evolution [1,3,71]. Our comparative analysis, supported by functional enrichment findings, indicates that while conventional EoE is strongly associated with epithelial barrier dysfunction and extracellular matrix remodeling, EoE-like esophagitis is predominantly immune-mediated, marked by the involvement of immunoglobulin complexes, antigen-binding activity, and the activation of CXCR3 chemokine pathways (*CXCL9, CXCL10, CXCL11*) facilitating systemic T-cell recruitment. The detection of blood microparticles further implies broader systemic immune activation, reinforcing the hypothesis that EoE-like esophagitis is driven by a dysregulated immune response rather than localized epithelial barrier compromise [1]. These findings have significant implications for therapeutic interventions, where early immunomodulation may prevent disease progression to classic EoE.

## Lymphocytic esophagitis: Neuro-immune dysregulation and metabolic dysregulation

Our current understanding of lymphocytic esophagitis' pathophysiology remains limited, with its etiology largely unknown. Lymphocytic esophagitis is characterized by intraepithelial lymphocyte infiltration, but specific immune pathways, such as interferon signaling, have not been well-defined in its context. Most existing studies focus on associations with conditions like GERD, motility disorders, and autoimmune phenomena, without pinpointing interferon involvement. The upregulation of *STAT1, CXCL10, and IRF1* underscores a strong interferon-mediated immune response, further differentiating it from the Th2-driven inflammation seen in conventional EoE [72]. These findings align with growing evidence that lymphocytic esophagitis may share immunopathogenic features with autoimmune disorders, warranting further investigation into its overlap with conditions such as celiac disease and Crohn's disease [73]. Another remarkable finding from this study is the strong association between lymphocytic esophagitis and mitochondrial dysfunction, as indicated by the downregulation of oxidative phosphorylation genes (*NDUFB2, NDUFS3, COX7A2, ATP5F1B, ATP5PB*, and *NDUFA2*) and enrichment in neurodevelopmental pathways. This suggests that lymphocytic esophagitis may involve metabolic dysregulation affecting esophageal sensory function and motility, potentially linking it to fatigue and systemic symptoms.

## Nonspecific esophagitis: A humoral immune-dominated subtype

Unlike EoE and EoE-like esophagitis, nonspecific esophagitis is strongly associated with humoral immunity, characterized by immunoglobulin complex enrichment and antigen-binding activity. The identification of *IgG, IgM,* and *IgA* pathways suggests a B-cell-driven immune response, distinct from the T-cell-dominated profiles of other esophagitis subtypes. This aligns with studies suggesting that nonspecific esophagitis may arise from chronic antigen exposure, food allergens, or systemic immune dysregulation. The systemic immune involvement inferred from blood microparticle enrichment suggests that nonspecific esophagitis may also have an autoimmune component, differentiating it from conventional EoE.

## Comparative mechanistic synthesis across esophagitis subtypes

Our integrative analyses indicate that across the subtypes, a shared immune–epithelial axis emerges, but with distinct emphases. Conventional EoE is Th2-driven with barrier disruption and ECM remodeling, while EoE-like shows stronger innate activity and milder barrier changes. Lymphocytic esophagitis is marked by interferon-chemokine programs and metabolic stress, and nonspecific esophagitis by humoral signatures, supporting a spectrum model shaped by cytokine polarity, epithelial integrity, and metabolic state. Random Forest classification reinforced these distinctions, with a minimal gene panel accurately separating conventional EoE from lymphocytic esophagitis.

## Transcription factor pathways in EoE and its subtypes

Notably, transcription factor activity (e.g., *FOXA1, STAT3, SMAD3, STAT1/IRF1*) reinforced the subtype-specific patterns observed in DEG and WGCNA analyses, supporting a unifying regulatory axis that connects immune polarization, epithelial remodeling, and metabolic stress across esophagitis subtypes.

Transcription factors play a crucial role in modulating gene expression pathways that drive the inflammatory and metabolic landscapes of eosinophilic esophagitis (EoE) and its subtypes, including EoE-like esophagitis, lymphocytic esophagitis, and nonspecific esophagitis. This study provides novel insights into disease-specific transcriptional networks, identifying key regulators that may contribute to immune activation, epithelial remodeling, and metabolic dysfunction across these esophageal conditions.

In analysis of most highly enriched TF pathways, *FOXA*1 emerged as a master regulator across multiple EoE subtypes, which cross-talks with both *STAT3* and *CREB1*-driven pathways. *FOXA1* is known to modulate epithelial differentiation and immune responses, suggesting that its dysregulation may contribute to barrier dysfunction and chronic inflammation in these conditions [74] . The interplay between *FOXA1, STAT3*, and *CREB1* indicates a coordinated regulatory mechanism where epithelial integrity is compromised by inflammatory cytokine signaling, leading to persistent esophageal inflammation and remodeling (Yu et al., 2014). These findings align with previous studies demonstrating *STAT3* involvement in chronic inflammatory diseases, particularly in mediating IL-13-driven epithelial barrier dysfunction [52].

A variety of distinct TF pathways differentiated conventional EoE from its subtypes. In lymphocytic esophagitis, a notable upregulation of *STAT1, IRF1*, and *CXCL10* suggests a strong interferon-mediated immune response, reinforcing its potential autoimmune-like characteristics. The enrichment of *STAT2* and *IRF1* further supports the hypothesis that lymphocytic esophagitis shares immunopathogenic features with Th1-driven inflammatory conditions, such as Crohn's disease and celiac disease [73]. In contrast, EoE-like esophagitis exhibited a predominantly immune-activated state, with CXCR3 ligand activation (*CXCL9, CXCL10, CXCL11*) and *FOXA1*-regulated pathways suggesting an adaptive immune response preceding eosinophilic infiltration. This immune priming phase may explain why some EoE-like patients progress to conventional EoE, while others exhibit persistent immune dysregulation without significant eosinophilic inflammation [51].

Another key finding was the presence of *RORB* and *NR4A1* transcription factor activity, which exhibited condition-specific roles in EoE and lymphocytic esophagitis, respectively. While *RORB* was associated with neuro-immune signaling, potentially influencing esophageal motility and sensory dysfunction in lymphocytic esophagitis, *NR4A1* contributed to epithelial remodeling and inflammatory cell recruitment in conventional EoE. These findings suggest that transcription factor regulation in esophageal inflammation extends beyond immune activation, potentially influencing neuro-modulatory responses and tissue homeostasis [75] .

In nonspecific esophagitis, transcriptional activity associated with immunoglobulin complexes and B-cell activation points to a fundamentally distinct pathophysiology, driven by humoral immune mechanisms rather than cellular inflammation. The role of B-cell-related transcription factors, such as *PAX5* and *IRF4*, in driving antigen presentation and antibody production, suggests that nonspecific esophagitis may be more closely linked to systemic immune dysregulation rather than localized eosinophilic inflammation. This differentiation from conventional EoE is critical for refining diagnostic criteria and identifying patients who may benefit from immunomodulatory therapies targeting B-cell activity [76].

Overall, our findings also demonstrate the molecular heterogeneity of esophageal inflammatory disorders. Each subtype—conventional EoE, EoE-like, lymphocytic, and nonspecific esophagitis—exhibited distinct transcriptional and regulatory patterns, particularly in pathways governing immune activation, epithelial barrier integrity, and stress response. While conventional EoE was characterized by strong Th2 cytokine signatures, lymphocytic and nonspecific esophagitis displayed more variable immune networks, with enrichment of stress-related and metabolic pathways. EoE-like esophagitis further diverged by showing incomplete activation of canonical Th2-associated programs. These differences suggest that, although the disorders share overlapping immunopathogenic frameworks, they diverge at the molecular level in ways that likely influence clinical presentation and therapeutic response.

## Implications for disease classification and treatment

Our findings suggest that EoE-like esophagitis exists along a disease spectrum with conventional EoE, with immune dysregulation preceding epithelial damage. In contrast, lymphocytic esophagitis and nonspecific esophagitis represent distinct

pathological entities, with metabolic function and humoral immunity as defining characteristics, respectively. These molecular distinctions provide a framework for subtype-specific diagnostic and therapeutic strategies, emphasizing the need for precision medicine approaches in esophageal disease management. Nevertheless, longitudinal studies are needed to elucidate the precise immunological and physiological mechanisms underpinning the progression from EoE-like subtype to conventional EoE and vice versa, particularly conditions contributing to epithelial barrier dysfunction.

To illustrate how these findings could be translated into a practical diagnostic context, we propose a tiered diagnostic schematic that integrates histopathologic evaluation, immune-axis profiling, and subtype-specific molecular markers (S7 Fig). This framework proposes the use of candidate biomarkers such as *POSTN, DNAH11*, *CXCL10*, and *STAT1* to complement current histologic criteria to improve subtype differentiation and diagnostic accuracy. While this schematic offers a promising conceptual model, it should be regarded as hypothesis-generating, requiring validation in prospective clinical studies before any diagnostic application.

## Study limitations

Despite the strengths of our integrative multi-omics approach and robust machine learning framework, several limitations should be noted. First, the sample size, particularly for some subtypes such as lymphocytic and nonspecific esophagitis, was relatively small, which may affect the generalizability of the findings and limit the statistical power to detect subtler molecular changes. Second, while transcriptomic analyses provide valuable insights into gene expression dynamics, they do not capture post-transcriptional or translational modifications that may influence protein activity and immune responses. Third, the cross-sectional nature of the dataset limits our ability to infer temporal progression or causality, particularly in the hypothesized transition from EoE-like to conventional EoE. Lastly, although we used established regulons and TF activity inference tools, experimental validation of key transcription factors and biomarkers (for example, *POSTN, DNAH11, CDX2*) in independent cohorts is warranted to confirm their diagnostic and therapeutic potential. Addressing these limitations in future longitudinal and multi-center studies will be critical for refining our understanding of EoE subtypes and their clinical implications. While independent validation datasets for all esophagitis subtypes are currently unavailable, internal cross-validation within GSE148381 (via WGCNA, TF inference, and Random Forest modeling) provided convergent evidence supporting the robustness of identified subtype-specific markers.

## Overall conclusion

Overall, this study supports the characterization of EoE-like esophagitis as an early, systemic immune response that, in some cases, progresses to a more localized and intensive inflammatory disorder in the form of classic EoE, highlighting both subtypes as part of a disease spectrum and refining our understanding of esophageal immune pathophysiology. The molecular and immunological distinctions identified between conventional EoE and its subtypes—together with the proposed diagnostic framework (S7 Fig)— provide a foundation for refining diagnostic criteria, redefining disease classification, and guiding the development of targeted therapeutic strategies, including biomarker-driven diagnostics, immunomodulatory interventions, and metabolic therapies.

## Supporting information

**S1 Fig. Top 10 DEGs between healthy controls and patients with (A) Conventional EoE (B) EoE-like esophagitis (C) Lymphocytic esophagitis (D) Nonspecific esophagitis.** The results highlight key genes *POSTN1* as one of the key genes differentially upregulated in conventional EoE patients.
(PPTX)

**S2 Fig. (A) Clustering of module eigengenes shows hierarchical relationships between gene modules, (B) Heatmap representation of all identified gene modules, illustrating their co-expression patterns, (C) Correlation**

between module membership and gene significance for the white module in Conventional EoE, highlighting hub genes potentially involved in disease progression, and (D) Similar analysis for the white module in EoE-like esophagitis, identifying key molecular players unique to this subtype.
(PPTX)

**S3 Fig. Determination of the optimal number of clusters using (A) the Elbow Method and (B) Silhouette Analysis.** The Elbow Method plots the within-cluster sum of squares (WCSS) against the number of clusters, with the "elbow" point indicating the optimal k. The Silhouette Analysis evaluates cluster separation, with higher silhouette scores reflecting better-defined clusters. Both methods suggested that three clusters provide the best representation of the dataset.
(PPTX)

**S4 Fig. Transcriptional regulation inferred for (A) Conventional EoE, (B) EoE-like esophagitis, (C) Lymphocytic esophagitis, and (D) Nonspecific esophagitis.** The analysis identifies key transcription factors driving gene expression changes in each subtype, shedding light on potential regulatory networks contributing to disease pathology.
(PPTX)

**S5 Fig. GO pathway of barrier-related genes in conventional and EoE-like esophagitis.** Red nodes represent the GO terms (specific cellular components) enriched in Conventional EoE, while the pink nodes represent the associated genes linked to these GO terms. This is consistent with the findings that EoE-like esophagitis does not exhibit significant structural barrier dysfunction compared to Conventional EoE, which showed strong evidence of downregulation in components like the cornified envelope (GO:0001533) and associated genes (e.g., *DSP, PKP3, TGM1, SCEL, SPRR1B, FLG*).
(PPTX)

**S6 Fig. Towards an immunological diagnostic framework for EoE and its subtypes: This schematic outlines a proposed tiered diagnostic framework integrating histological findings with immune-axis and molecular profiling to improve subtype differentiation in eosinophilic esophagitis (EoE) and related esophagitis phenotypes.** Tier 1 applies conventional histologic assessment, using the presence of ≥15 eosinophils per high-power field as the initial criterion for confirming conventional EoE. Tier 2 incorporates immune-axis profiling to identify Th2-dominant inflammation (↑IL4, IL5, IL13; ↓MT1X, MT2A), supporting the diagnosis of conventional EoE even in borderline eosinophil counts. Tier 3 extends classification to include interferon-driven (STAT1, IRF1, CXCL10) or mixed Th1/Th2 (CXCR3 ligands) signatures, as well as humoral-immune enrichment (IGHV, IGKV) to distinguish lymphocytic, EoE-like, and nonspecific esophagitis subtypes. Candidate biomarkers such as POSTN, DNAH11, CXCL10, STAT1, and immunoglobulin-related transcripts (IGHV, IGKV) represent potential molecular markers for future clinical validation. This conceptual flowchart illustrates how the study's subtype-specific molecular insights could be adapted into a diagnostic algorithm that bridges transcriptomic research with clinical decision-making.
(PPTX)

**S7 Fig. Immunological and molecular heterogeneity among esophagitis subtypes.** Simplified schematic summarizing the dominant immune pathways and molecular features identified for each subtype of esophagitis. Conventional EoE shows **Th2-driven inflammation**, zinc pathway suppression, and epithelial remodeling; **EoE-like esophagitis** exhibits a mixed **Th1/Th2 immune profile** and CXCR3-ligand signaling; **lymphocytic esophagitis** demonstrates **interferon-mediated Th1 activation** with mitochondrial and neuro-immune dysfunction; and **nonspecific esophagitis** is characterized by **humoral immune activation** with strong B-cell and immunoglobulin gene enrichment.
(PPTX)

## Author contributions

**Conceptualization:** Aman Ullah, Eric Twum, Ancha Baranova.

**Data curation:** Eric Twum.

**Formal analysis:** Eric Twum.

**Investigation:** Aman Ullah, Eric Twum.

**Methodology:** Eric Twum.

**Project administration:** Aman Ullah, Ancha Baranova.

**Resources:** Aman Ullah.

**Software:** Eric Twum.

**Supervision:** Aman Ullah, Ancha Baranova.

**Visualization:** Eric Twum.

**Writing – original draft:** Eric Twum.

**Writing – review & editing:** Aman Ullah, Ancha Baranova.

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
