## [Decision Letter · Decision Letter 0]

12 Jul 2025

PONE-D-25-23211Molecular and Immunological Heterogeneity of Eosinophilic Esophagitis: Insights and SubtypingPLOS ONE

Dear Dr. Ullah,

Thank you for submitting your manuscript to PLOS ONE. After careful consideration, we feel that it has merit but does not fully meet PLOS ONE’s publication criteria as it currently stands. Therefore, we invite you to submit a revised version of the manuscript that addresses the points raised during the review process.

We look forward to receiving your revised manuscript.

Kind regards,

Wan-Tien Chiang

Academic Editor

PLOS ONE

Journal Requirements:

6. Please upload a new copy of Figure 1, 2, 3, 4, 5, 6, 7, and 8 as the detail is not clear. Please follow the link for more information: https://blogs.plos.org/plos/2019/06/looking-good-tips-for-creating-your-plos-figures-graphics/" https://blogs.plos.org/plos/2019/06/looking-good-tips-for-creating-your-plos-figures-graphics/"

Reviewers' comments:

Reviewer's Responses to Questions

**Comments to the Author**

1. Is the manuscript technically sound, and do the data support the conclusions?

Reviewer #1: Yes

Reviewer #2: Yes

Reviewer #3: Yes

Reviewer #4: Yes

2. Has the statistical analysis been performed appropriately and rigorously? 

Reviewer #1: Yes

Reviewer #2: Yes

Reviewer #3: Yes

Reviewer #4: Yes

3. Have the authors made all data underlying the findings in their manuscript fully available?

Reviewer #1: Yes

Reviewer #2: Yes

Reviewer #3: Yes

Reviewer #4: Yes

4. Is the manuscript presented in an intelligible fashion and written in standard English?

Reviewer #1: No

Reviewer #2: Yes

Reviewer #3: Yes

Reviewer #4: Yes

5. Review Comments to the Author

Reviewer #1: > Overall Assessment:

The manuscript is well-written and presents a significant contribution to the field. It is generally acceptable; however, some revisions are necessary to enhance its readability, clarity, and adherence to journal standards. Below are specific comments to assist in improving the manuscript:

> Title:

The title is clear, concise, and accurately reflects the content of the manuscript. No changes are necessary.

> Abstract:

The abstract is well-written and effectively conveys the study's objective, methodology, and findings.

The state-of-the-art nature of the study is also clear.

Suggestion: Ensure that the keywords adequately represent the study and enhance its discoverability in relevant research areas.

> Discussion:

While the discussion section provides valuable insights, it lacks a detailed examination of the important topics and implications of the study.

Recommendation: Include a systematic discussion of the study's limitations to provide a balanced perspective and guide future research in this area.

> Conclusion:

The conclusion effectively summarizes the study's findings and their significance. No major revisions are necessary in this section.

> References:

Several references do not adhere to the journal's formatting guidelines.

Recommendation: Carefully check and update all references to conform to the required style, including journal abbreviations and citation formats.

Avoid citing outdated references; instead, prioritize high-quality, relevant publications, particularly those from 2023, to strengthen the manuscript's credibility and relevance.

> Language and Readability

Although the manuscript is generally well-written, numerous sentences contain minor typographical and grammatical errors.

Suggestion: Utilize a professional English editing service to refine the language and improve overall readability.

Reviewer #2: Dear Author

Thank you for your manuscript submission. The current manuscript is well-designed and well-presented. However, a Minor Revision is needed as below:

1. As the current work is an in silico one, please do add the terms of "in silico" and "Dry Lab" to Methodology section. In this regard, the following paper is a useful reference for the current manuscript:

Respiratory Tularemia: Francisella Tularensis and Microarray Probe Designing. Open Microbiol J. 2016 Nov 30;10:176-182. doi: 10.2174/1874285801610010176. PMID: 28077973; PMCID: PMC5204066.

2. Please do add a flow chart to Methodology section to show all the procedures done within the current study.

3. It is recommended to add a flow chart to Results section to show the main outcomes obtained from the current study.

4. Please do add the strength and the limitation of the current study.

5. Please do interpret the results of the statistical analyses within the current study.

Reviewer #3: Introduction:

1. For conventional eosinophilic esophagitis (EoE) and its subtypes—EoE-like esophagitis, lymphocytic esophagitis, and nonspecific esophagitis, can authors demonstrate what are common features of immunology, genetics, and epigenetics?

2. How do findings show molecular heterogeneity of esophageal inflammatory disorders? Can authors make more detailed descriptions?

Methods:

1. For Figure 1, do TF analysis, WGCNA, DE analysis and ML analysis share a parallel relationships? If not, can authors modify this figure?

2. For Differential Gene Expression, why do authors select “Genes exhibiting |log2 fold change (FC)| > 1.5 and a p-adjusted value of p < 0.05 were considered statistically significant”?

3. For Gene Ontology and Functional Analysis, if it is a kind of ORA analysis, authors should indicate this point?

4. Do Dorothea and WGCNA share similar application conditions?

5. “gene lists from the modules of interest into the STRING”, do they come from WGCNA analysis?

RESULTS:

1. Explain why are belt shape for A and C are different from B and D in Figure 2?

2. “GO-BP revealed distinct clustering patterns and varying correlation levels”, can authors explain by disease mechanisms research progress?

3. What are common/different findings between DE-GO analysis and WGCNA-GO analysis?

4. How are hub genes identified in PPI network by WGCNA module related with findings from DE analysis?

5. Have authors try unsupervised learning?

6. Can results of TF analysis explain the results in DE/WGCNA analysis?

DISCUSSION

1. How are the immune mechanisms in EoE related with ECM remodeling, and cell motility and organ morphogenesis,

especially for Th2-mediated inflammation?

2. How are Epigenetic modifications involved in EoE? Are they connected to immune?

3. Why is EoE-like esophagitis are different from cEoE, by mechanisms? How are about Lymphocytic Esophagitis, Nonspecific Esophagitis? Can authors link them together using your findings?

4. For TF results, again, can authors link them together using your findings?

Reviewer #4: The submitted manuscript presents a comprehensive and insightful multi-omics investigation into the molecular and immunological heterogeneity of eosinophilic esophagitis (EoE) and its subtypes. The integrative approach utilizing differential gene expression profiling, weighted gene co-expression network analysis (WGCNA), and machine learning techniques is robust and methodologically sound.

Strengths:

- The study addresses an important diagnostic gap in distinguishing between EoE subtypes.

- Identification of subtype-specific biomarkers such as DNAH11, CXCL10, and zinc transporters (SLC39A1, SLC39A2) is novel and therapeutically promising.

- The inclusion of both metabolic and immune-related signatures adds depth to the mechanistic understanding of each subtype.

- The manuscript is well-organized and clearly written, with logical progression from background to results and implications.

Minor Revisions Suggested:

- While the overall analysis is clearly outlined, the authors should provide more detail on the machine learning models used—especially the type, validation strategy, and performance metrics (e.g., accuracy, AUC).

- If any qPCR or external validation datasets were used to confirm key findings (e.g., POSTN, DNAH11 expression), they should be briefly described. If not performed, suggest acknowledging this as a limitation.

- Although the statement claims full availability, including a link to the raw and processed gene expression data repository (e.g., GEO accession number) would improve transparency.

- Consider adding a visual summary or schematic figure outlining the subtype-specific molecular mechanisms for easier reader interpretation.

6. PLOS authors have the option to publish the peer review history of their article (what does this mean? ). If published, this will include your full peer review and any attached files.

**Do you want your identity to be public for this peer review?** For information about this choice, including consent withdrawal, please see our Privacy Policy .

Reviewer #1: No

Reviewer #2: **Yes:** Payam BEHZADI

Reviewer #3: No

Reviewer #4: No

---

## [Author Response · Author response to Decision Letter 1]

23 Aug 2025

I have incorporated all the changes recommended by the reviewers in the uploaded document. All modifications are highlighted for your convenience.

Please let me know if you would prefer a clean version of the manuscript without highlights—I’d be happy to provide it.

---

## [Decision Letter · Decision Letter 1]

23 Sep 2025

PONE-D-25-23211R1Molecular and Immunological Heterogeneity of Eosinophilic Esophagitis: Insights and SubtypingPLOS ONE

Dear Dr. Ullah,

Thank you for submitting your manuscript to PLOS ONE. After careful consideration, we feel that it has merit but does not fully meet PLOS ONE’s publication criteria as it currently stands. Therefore, we invite you to submit a revised version of the manuscript that addresses the points raised during the review process.

We look forward to receiving your revised manuscript.

Kind regards,

Austin W.T. Chiang

Academic Editor

PLOS ONE

Journal Requirements:

Reviewers' comments:

Reviewer's Responses to Questions

**Comments to the Author**

1. If the authors have adequately addressed your comments raised in a previous round of review and you feel that this manuscript is now acceptable for publication, you may indicate that here to bypass the “Comments to the Author” section, enter your conflict of interest statement in the “Confidential to Editor” section, and submit your "Accept" recommendation.

Reviewer #1: All comments have been addressed

Reviewer #2: (No Response)

Reviewer #4: (No Response)

2. Is the manuscript technically sound, and do the data support the conclusions?

Reviewer #1: Yes

Reviewer #2: (No Response)

Reviewer #4: Yes

3. Has the statistical analysis been performed appropriately and rigorously? 

Reviewer #1: Yes

Reviewer #2: (No Response)

Reviewer #4: Yes

4. Have the authors made all data underlying the findings in their manuscript fully available?

Reviewer #1: Yes

Reviewer #2: (No Response)

Reviewer #4: Yes

5. Is the manuscript presented in an intelligible fashion and written in standard English?

Reviewer #1: Yes

Reviewer #2: (No Response)

Reviewer #4: Yes

6. Review Comments to the Author

Reviewer #1: The manuscript is well-written, clearly structured, and presents original research findings that are relevant to the field. The study design, methodology, data analysis, and interpretation are appropriate and thorough. Figures and tables are well-prepared, and the discussion accurately reflects the findings while placing them in context with prior literature. The references are comprehensive and up-to-date, and the conclusions are well-supported by the data.

I found no major issues related to dual publication, plagiarism, or research ethics. The authors appear to have followed proper ethical standards. No concerns regarding publication ethics were identified.

Recommendation: Accept the manuscript in its current form.

Reviewer #2: (No Response)

Reviewer #4: The manuscript “Molecular and Immunological Heterogeneity of Eosinophilic Esophagitis: Insights and Subtyping” provides a comprehensive integrative multi-omics and bioinformatics approach to disentangling the heterogeneity of EoE and its related subtypes. The study is timely and addresses an important unmet need in gastroenterology and immunology, namely the lack of clear molecular criteria to differentiate overlapping esophageal inflammatory disorders.

Strengths:

- The integrative approach using gene expression profiling, WGCNA, enrichment studies, and machine learning provides robust cross-validation of findings.

- Identification of novel players such as DNAH11 and zinc pathway alterations adds originality and translational relevance.

- The clear delineation of subtype-specific pathways (e.g., immune dysregulation in EoE-like, mitochondrial impairment in lymphocytic esophagitis) is of high significance for potential precision medicine.

- The manuscript is generally well-written and logically structured.

Points for Improvement:

- While analyses appear rigorous, the manuscript would benefit from clearer reporting of sample sizes, adjusted p-values, and validation across independent datasets where possible. This will strengthen confidence in subtype-specific conclusions.

- Pathway maps or schematic diagrams summarizing the distinct molecular landscapes of each subtype would enhance clarity for readers less familiar with systems biology.

- The connection between altered zinc homeostasis and epithelial dysfunction is compelling but currently speculative. Additional evidence or literature support should be added to avoid overinterpretation.

- Minor grammatical edits are needed (e.g., in the Results section, a few sentences are overly long and could be broken down for clarity).

- The conclusion should emphasize how these findings could realistically be applied in clinical diagnostics or therapeutic development. A short section outlining potential biomarkers suitable for clinical validation would be valuable.

This manuscript is technically sound, presents novel insights, and makes a meaningful contribution to the understanding of EoE heterogeneity. Pending minor revisions (clarity, reporting detail, and expanded discussion on translational relevance), it is suitable for publication.

7. PLOS authors have the option to publish the peer review history of their article (what does this mean? ). If published, this will include your full peer review and any attached files.

**Do you want your identity to be public for this peer review?** For information about this choice, including consent withdrawal, please see our Privacy Policy .

Reviewer #1: No

Reviewer #2: **Yes:** Payam BEHZADI

Reviewer #4: No

---

## [Author Response · Author response to Decision Letter 2]

22 Oct 2025

We have carefully addressed all the comments provided by the reviewers and the editor and revised the manuscript accordingly. A detailed, point-by-point response to each comment has been included, along with marked changes in the revised manuscript. Additionally, we have added supplementary figures as requested by the reviewers.

---

## [Decision Letter · Decision Letter 2]

13 Jan 2026

PONE-D-25-23211R2Molecular and Immunological Heterogeneity of Eosinophilic Esophagitis: Insights and SubtypingPLOS One

Dear Dr.  Man Ullah,

Thank you for submitting your manuscript to PLOS ONE. After careful consideration, we feel that it has merit but does not fully meet PLOS ONE’s publication criteria as it currently stands. Therefore, we invite you to submit a revised version of the manuscript that addresses the points raised during the review process. ==================================

Number 3 of the POS One publications criteria required that "Experiments, statistics, and other analyses are performed to a high technical standard and are described in sufficient detail."

While the analyses appear rigorous, the manuscript should include clear reporting of sample sizes, adjusted p-values, and validation across independent datasets where possible. 

To address no. 4 "Conclusions are presented in an appropriate fashion and are supported by the data.", please include additional literature for the connection between zinc homeostasis and epithelial dysfunction.

5. is that  "The article is presented in an intelligible fashion and is written in standard English."

Minor grammatical edits are needed (e.g., in the Results section, a few sentences are overly long and could be broken down for clarity).

We look forward to receiving your revised manuscript.

Kind regards,

Claudia D. Andl, Ph.D.

Academic Editor

PLOS One

Journal Requirements:

Additional Editor Comments (if provided):

While the reviewers overall agree that this manuscript is of interest and technically sound, it would benefit from clearer reporting of sample sizes, adjusted p-values, and validation across independent datasets where possible to support the subtype-specific conclusions.

Similarly, to enhance the connection between altered zinc homeostasis and epithelial dysfunction, additional evidence from the literature should be added to avoid overinterpretation.

Minor grammatical edits are needed (e.g., in the Results section, a few sentences are overly long and could be broken down for clarity).

Reviewers' comments:

Reviewer's Responses to Questions

**Comments to the Author**

1. If the authors have adequately addressed your comments raised in a previous round of review and you feel that this manuscript is now acceptable for publication, you may indicate that here to bypass the “Comments to the Author” section, enter your conflict of interest statement in the “Confidential to Editor” section, and submit your "Accept" recommendation.

Reviewer #2: All comments have been addressed

2. Is the manuscript technically sound, and do the data support the conclusions?

Reviewer #2: Yes

3. Has the statistical analysis been performed appropriately and rigorously? 

Reviewer #2: Yes

4. Have the authors made all data underlying the findings in their manuscript fully available?

Reviewer #2: Yes

5. Is the manuscript presented in an intelligible fashion and written in standard English?

Reviewer #2: Yes

6. Review Comments to the Author

Reviewer #2: Dear Author

Thank you for your effective revision. Hence, the current manuscript can be published in present form

7. PLOS authors have the option to publish the peer review history of their article (what does this mean? ). If published, this will include your full peer review and any attached files.

**Do you want your identity to be public for this peer review?** For information about this choice, including consent withdrawal, please see our Privacy Policy .

Reviewer #2: **Yes:** Payam BEHZADI

---

## [Author Response · Author response to Decision Letter 3]

26 Jan 2026

Please find below our point-by-point responses to the Editor’s and Reviewers’ comments, including the corrected response to Reviewer #4.

---

## [Editor Report · Decision Letter 3]

29 Jan 2026

Molecular and Immunological Heterogeneity of Eosinophilic Esophagitis: Insights and Subtyping

PONE-D-25-23211R3

Dear Dr. Aman Ullah,

We’re pleased to inform you that your manuscript has been judged scientifically suitable for publication and will be formally accepted for publication once it meets all outstanding technical requirements.

Kind regards,

Claudia D. Andl, Ph.D.

Academic Editor

PLOS One
---

## [Editor Report · Acceptance letter]

PONE-D-25-23211R3

PLOS One

Dear Dr. Ullah,

I'm pleased to inform you that your manuscript has been deemed suitable for publication in PLOS One. Congratulations! Your manuscript is now being handed over to our production team.

Kind regards,

on behalf of

Dr. Claudia D. Andl

Academic Editor

PLOS One